# Human detection of political speech deepfakes across transcripts, audio, and video

Matthew Groh [1,4] ✉, Aruna Sankaranarayanan[2,3,4], Nikhil Singh [2], Dong Young Kim [2], Andrew Lippman[2] & Rosalind Picard [2]

Recent advances in technology for hyper-realistic visual and audio effects provoke the concern that deepfake videos of political speeches will soon be indistinguishable from authentic video. We conduct 5 pre-registered randomized experiments with N = 2215 participants to evaluate how accurately humans distinguish real political speeches from fabrications across base rates of misinformation, audio sources, question framings with and without priming, and media modalities. We do not find base rates of misinformation have statistically significant effects on discernment. We find deepfakes with audio produced by the state-of-the-art text-to-speech algorithms are harder to discern than the same deepfakes with voice actor audio. Moreover across all experiments and question framings, we find audio and visual information enables more accurate discernment than text alone: human discernment relies more on how something is said, the audio-visual cues, than what is said, the speech content.

Recent advances in technology for algorithmically applying hyper-realistic manipulations to video are simultaneously enabling new forms of interpersonal communication and posing a threat to traditional standards of evidence and trust in media[1–8]. In the last few years, computer scientists have trained machine learning models to generate photorealistic images of people who do not exist[9–12], inpaint people out of images[13,14], clone voices based on a few samples[15,16], modulate the lip movements of people in videos to make them appear to say something they have not said[17,18], and create synthetic videos based on simple text prompts[19]. The synthetic videos' false appearance of indexicality – the presence of a direct relationship between the photographed scene and reality[20,21] – has the potential to lead people to believe video-based messages that they otherwise would not have believed if the messages were communicated via text. This potential influence is particularly concerning because research demonstrates that videos, especially videos of an injustice, elicit more engagement and emotional reactions (e.g., anger, sympathy) than text descriptions displaying the same information[22–24] (although, see ref. 25). Moreover,

visual misinformation appears on social media[26] (although, see ref. 27 documenting a pattern of low exposure of people in general to misinformation) and the emotional and motivational influences of visual communication have been attributed to why misleading viral videos have provoked mob-violence[28,29]. While people are more likely to believe a real event occurred after watching a video of the event than reading a description of the event[30], an open question remains: Does visual communication relative to text or audio increase the believability of fabricated events?

The realism heuristic[29,31] predicts that "people are more likely to trust audiovisual modality [relative to text] because its content has a higher resemblance to the real world." This prediction is relevant for many deepfake videos[32] and suggests fabricated video would be more believable than fabricated text conditional on the absence of obvious perceptual distortions. Yet there exists little direct empirical evidence for this heuristic applied to algorithmically manipulated video. In an experiment using 3 false news videos as stimuli, researchers found that stories presented as videos are perceived as more credible than stories

[1]Kellogg School of Management, Northwestern University, Evanston, IL, USA. [2]Media Lab, Massachusetts Institute of Technology, Cambridge, MA, USA. [3]CSAIL, Massachusetts Institute of Technology, Cambridge, MA, USA. [4]These authors contributed equally: Matthew Groh, Aruna Sankaranarayanan. ✉e-mail: matthew.groh@kellogg.northwestern.edu

presented as text or read aloud in audio form[29]. In contrast, in an experiment by Barari et al. 2021 showing 6 political deepfake videos (videos manipulated by artificial intelligence to make someone say something they did not say) and 9 non-manipulated videos, researchers did not find differences between truth discernment rates in video, audio, and text[33]. Perhaps some of the participants did not take the videos' indexicality as evidence of authenticity because participants were aware of how easily such videos could be manipulated. Alternatively, some participants may have noticed perceptual distortions or indicators of satire (e.g., facial expressions and comedic timing) in the videos used in Barari et al. 2021, which would naturally lead one to believe the video has been manipulated. In experiments examining how people react to deepfake videos of politicians, researchers find people are not more likely to report false memories after watching deepfake videos than reading the same false news as text[34], people are more likely to feel uncertain than misled after viewing a deepfake of Barack Obama[35], people consider a deepfake of a Dutch politician significantly less credible than the real video from which it was adapted[36] and the previously mentioned deepfake video of the Dutch politician is not more persuasive than the text alone[37]. In the experiment examining the deepfake of the Dutch politician, some respondents explained their credibility judgments by indicating audio-visual cues of how the message was communicated (e.g., unnatural mouth movements); others indicated inconsistency in the content of the message itself (e.g., contextually unrealistic speeches)[36]. One explanation for the current mixed evidence on the role of communication modalities in mediating people's ability to discern fabricated content is the large possibility space of how political speeches may appear in videos and the small number of stimuli in media effects research[38].

Related research demonstrates how fake images can be persuasive and difficult to distinguish from real images. People rarely question the authenticity of images even when primed[39]. Images can increase the credibility of disinformation[40]. Images of synthetic faces produced by StyleGAN2[10] are indistinguishable from the original photos on which the StyleGAN2 algorithm was trained[41]. Moreover, research shows that non-probative and uninformative photos can lead people to believe false claims[42], lead people to believe they know more than they actually know[43], and promote truthiness by creating illusory truth effects[44,45], which can lead people to believe falsehoods they previously knew to be falsehoods[46,47]. When it comes to ostensibly probative videos of political speeches, the question of whether people are more likely to believe an event occurred because they saw it as opposed to only reading about it remains open.

In fact, today's algorithmically generated deepfakes are not yet consistently indistinguishable from real videos. On a sample of 166 videos from the largest publicly available dataset of deepfake videos to date from the Deepfake Detection Contest (DFDC)[48], people are significantly better than chance but far from perfect at discerning whether an unknown actor's face has been visually manipulated by a deepfake algorithm[49]. This finding is significant because it demonstrates that people can identify deepfake videos from real videos based solely on visual cues. However, some videos are more difficult than others to distinguish because of their blurry, dark, or grainy visual features. On a subset of 11 of the 166 videos, Kobis et al. 2021 do not find that people can detect deepfakes better than chance[50]. Likewise, Lovato et al. 2024 find that participants are only 51% accurate at identifying videos from the DFDC when first asked questions about the likability of the video content, and only after initially answering these questions are asked whether the primary person in the video was real or fictionally created[51].

In another experiment with 25 deepfake videos and 4 real videos but only 94 participants, researchers found that the overall discernment accuracy is 51%, and media literacy training increases discernment accuracy by 24 percentage points for participants assigned to the training relative to the control group[52].

People's capacity to identify multimedia manipulations raises questions: how do various kinds of fabricated media (e.g., synthesized audio and video of political speeches that never happened) alter the perceived credibility of misinformation, how do audience characteristics (e.g., reflective reasoning) moderate media effects, and how does the source and content of a message interact with the fabricated media and audience characteristics[53]? A growing field of misinformation science is beginning to address these questions. Research on news source quality demonstrates that people in the United States are generally accurate at identifying high and low-quality publishers[54] and the salience of source information does not appear to change how accurately people identify fabricated news stories[55], manipulated images[56], or false news headlines[57,58] although evidence on false news headlines is mixed[59,60]. Research on political false news content suggests an individual's tendency to rely on intuition instead of analytic thinking is a stronger factor than motivated reasoning in explaining why people fall for false news[61], and similarly, people with more analytic cognitive styles worldwide are more accurate at discerning between authentic and fabricated political videos[62] and true and false headlines related to COVID-19[63]. In fact, people tend to be better at discerning truth from falsehood when evaluating news headlines that are concordant with their political partisanship relative to when evaluating news headlines that are discordant[64]. While the science of misinformation has generally focused on the messengers (the source credibility of publishers)[65] and the message of what is said (the media credibility of written articles and headlines)[64], the relevance of audio-visual communication channels to the psychology of misinformation has received less attention[66] and is important for addressing the problem of misinformation[67].

In this paper, we conduct 5 pre-registered experiments to evaluate how well people can distinguish between real and fabricated political speeches by well-known politicians and how communication modalities, contexts, audio sources, base rates of fabricated content, and question framings influence discernment. The stimuli include 32 videos from the Presidential Deepfake Dataset[68] and 12 additional videos[33]. In total, we analyze data from 2215 recruited participants in these 5 pre-registered experiments and an additional 41,313 non-recruited participants who participated in Experiment 1 but were not pre-registered. Table 1 presents an overview of the five experiments.

We begin with Experiment 1, which addresses the question: How does media modality influence participants' ability to discern real and fabricated political speeches? In Experiment 1, we present participants with 32 political speeches – half real and half fabricated – by Donald Trump and Joseph Biden that are randomized to be displayed via the 7 possible permutations of text, audio, and video: a transcript, an audio clip, a silent video, audio with subtitles, silent video with subtitles, video with audio, and video with audio and subtitles. By randomly assigning political speeches to these permutations of text, audio, and video modalities and asking participants to discern truth from falsehood (see Methods section for exact question wording), this experiment is designed to disentangle the degree to which participants attend to and consider the content of what is said and the audio-visual cues as to how it is said.

Experiment 2 builds upon Experiment 1 by enhancing and extending the stimuli and adapting the wording of the experiment. In Experiment 2, we present participants with 20 videos randomly sampled from 60 videos of politicians, which include 12 videos used in Barari et al. 2021[33] and videos of the same 32 political speeches from Experiment 1 where the 16 real videos are the same and the deepfakes are enhanced with the state-of-the-art algorithms in 2023[69] and include 16 deepfakes with voice actor audio and 16 deepfakes with audio produced by a text-to-speech algorithm fine-tuned on the presidents' voices[70]. Experiment 2 offers an empirical investigation into human discernment of deepfake videos with different sources of audio (audio from a voice actor or text-to-speech algorithm) and different contexts

**Table 1 | Overview of the five experiments**

| Experiment | Prereg | Feedback | Known Rate | Stimuli | Obs | Modalities | Rate of Fabrications |
|---|---|---|---|---|---|---|---|
| Experiment 1a | Yes | Yes | Yes | 32 original PDD | 32 | Text, audio, and video | 50% |
| Experiment 1b | No | Yes | Yes | 32 original PDD | 32 | Text, audio, and video | 50% |
| Experiment 2 | Yes | No | No | 48 enhanced PDD and 12 other videos | 20 | Video | 60% |
| Experiment 3 | Yes | No | No | 32 enhanced PDD (TTS deepfakes) | 20 | Text, audio, and video | 20% or 80% |
| Experiment 4 | Yes | No | No | 16 real PDD videos | 16 | Audio and video | 50% |
| Experiment 5 | Yes | No | No | 32 enhanced PDD (TTS deepfakes) | 10 | Text, audio, and video | 50% |

Prereg indicates whether the experiment was pre-registered, Feedback indicates whether we give participants immediate feedback on whether a stimulus is fabricated or not, Known Rate indicates whether we informed participants the rate of fabrications, Stimuli refers to the stimuli used, Obs refers to the maximum number of observations provided by each participant, Modalities indicates the possible modalities in which the stimuli are presented, and Rate of Fabrications refers to the base rate of fabricated speeches, which was randomized in Experiment 3.

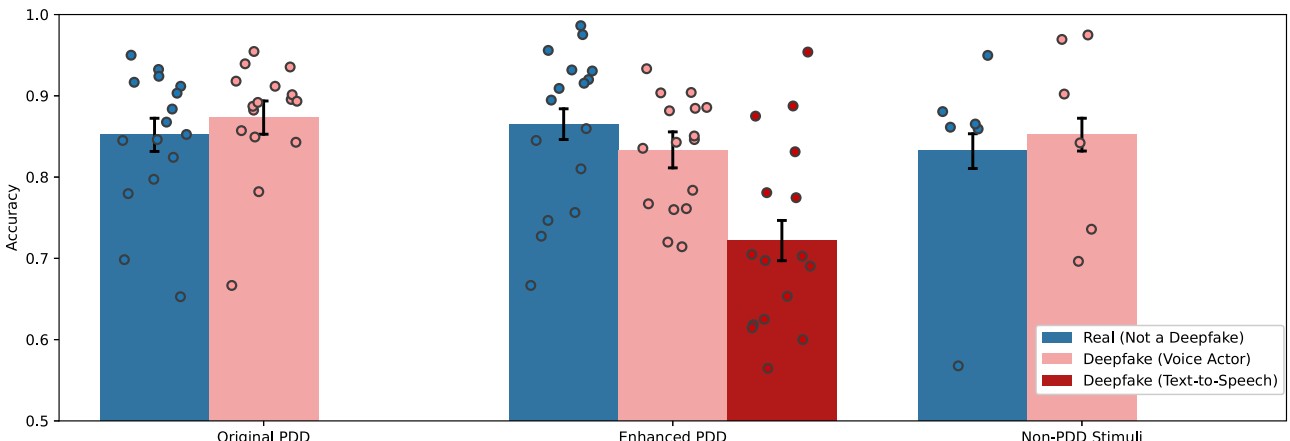

**Fig. 1 | Accuracy distinguishing real and fabricated speeches across video stimuli.** Accuracy across the original Presidential Deepfakes Dataset (PDD) video stimuli in Experiment 1a ($N = 2228$ observations), the enhanced PDD video stimuli in Experiment 2 ($N = 3580$ observations), and the non-PDD video stimuli in Experiment 2 ($N = 2384$ observations). The error bars represent 95% confidence intervals. Dot plots represent the mean accuracy for each video stimulus.

of videos (non-satirical presidential speeches in the Presidential Deepfake Dataset (PDD) videos and satirical speeches and explicit discussions of synthetic media in the other videos used in Barari et al. 2021).

In Experiment 3, we examine how the base rate of fabrications influences participant accuracy by randomizing participants to a low or high base rate of fabricated political speeches. We present participants with 20 political speeches sampled from 32 political speeches by Donald Trump and Joseph Biden that are randomized to appear as a transcript, a silent video, an audio clip, or a video with audio. Experiment 3 provides a conceptual replication of experiment 1 with enhanced stimuli and an opportunity to evaluate the influence of base rates of misinformation.

In Experiment 4, we present participants with 16 videos or audio clips of 16 real political speeches by Donald Trump and Joseph Biden and the same 16 real political speeches with audio produced by a voice actor; we ask participants if they can identify which stimuli are voiced by the US presidents and which are voiced by the voice actor. Experiment 4 offers an opportunity to evaluate how accurately participants can distinguish Donald Trump and Joseph Biden's voice from a voice actor's voice.

Finally, in Experiment 5, we present participants with 10 videos randomly sampled from the same 32 videos in Experiment 3, but we do not prime people with a direct question of authenticity. In contrast to the previous experiments, we asked participants, "What comes to mind after watching the following video/listening to the following audio/reading the following quote?" This final experiment reveals how even when participants are not necessarily paying attention to

authenticity they reveal suspicions of fabrications differently across media modalities.

## Results

Experiments 1a, 2, 3, 4, and 5 involve participants recruited from Prolific and are pre-registered on aspredicted.org at the following URLs: 1a, 2, 3, 4, and 5. Experiment 1b is not pre-registered and does not include any participant-level information beyond accuracy on stimuli, but it includes 41,313 participants who discovered the experiment organically through search engines or the news. In the Methods section, we provide details on participants, the digital experiment interface, experimental stimuli, and randomization protocol. Throughout this paper, accuracy refers to human participants, not machine learning models unless otherwise noted.

Figure 1 presents the accuracy of participants in Experiment 1a and Experiment 2 on real and fabricated videos with audio from the Presidential Deepfakes Data (PDD) videos and 12 other videos previously examined in Barari et al. 2021[33]. In Experiment 1a, participants correctly identified real PDD videos and deepfakes in 85% and 87% of observations, respectively. In Experiment 2, participants correctly identified real PDD videos, the enhanced PDD voice actor deepfakes, enhanced PDD text-to-speech deepfakes, real other videos, and other deepfakes in 86%, 83%, 72%, 85%, and 83% of observations. As a baseline for comparison, random guessing on this task would lead to 50% accuracy. Participants are closer to random guessing than a perfect score on the enhanced PDD text-to-speech deepfakes but closer to a perfect score on the rest of the stimuli.

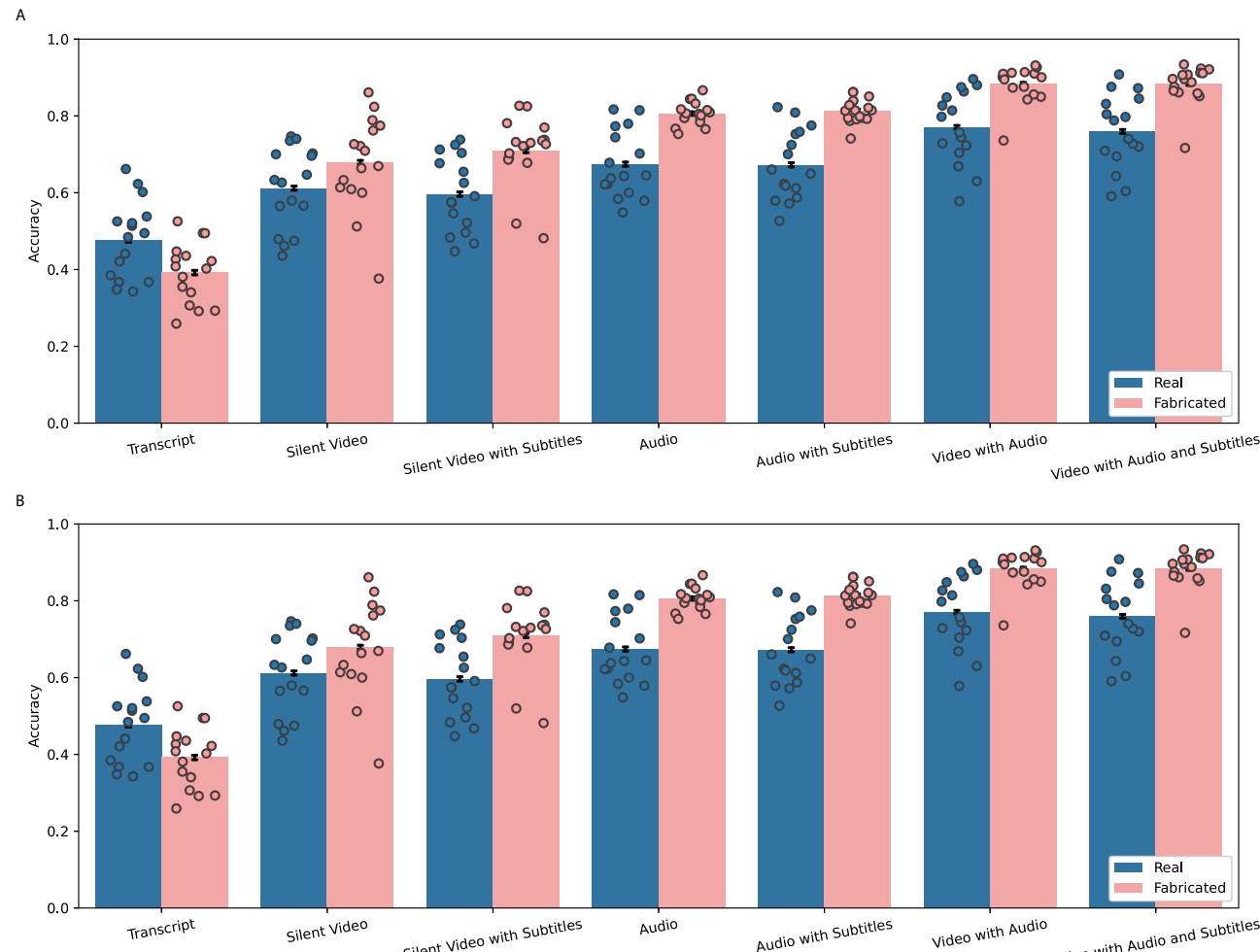

**Fig. 2 | Accuracy distinguishing real and fabricated speeches across media modalities in experiment 1. A** Mean accuracy across all permutations of text, audio, and video in Experiment 1a with 501 recruited participants ($N = 16{,}011$ observations). **B** Mean accuracy across all permutations of text, audio, and video in Experiment 1b with 41,313 non-recruited participants ($N = 416{,}901$ observations). The error bars represent 95% confidence intervals. Dot plots represent the mean accuracy for each video stimulus.

## Experiment 1a (501 participants, preregistered)

We designed Experiment 1a to address the following question: How does media modality influence participants' ability to discern real and fabricated political speeches? In order to answer this question, we show participants 32 videos from the original Presidential Deepfakes Dataset (PDD), inform participants that half are real and half are fake, and ask participants to indicate their level of confidence that the stimulus is a fabricated political speech or not. After each response, we inform participants of whether the stimulus was real or fabricated.

We find that participants' accuracy increases as they have access to additional communication modalities. In particular, accuracy by modality from lowest to highest starts with transcripts at 57% accuracy followed by silent videos without subtitles at 64% accuracy, silent videos with subtitles at 69% accuracy, audio (with and without subtitles) at 81% accuracy, video with audio and subtitles at 85% accuracy, and video with audio but no subtitles at 86% accuracy. Figure 2 shows accuracy across modalities for real and fabricated stimuli for Experiments 1a and 1b. In Supplementary Table 1, we present the pre-registered regression analysis on confidence score which is a measure of accuracy weighted by participants' confidence defined as the participant's confidence (ranging from 50 to 100) if correct and 100 minus the participant's confidence if incorrect. Based on ordinary least squares (OLS) regressions (all OLS regressions in this paper

include standard robust errors clustered at the participant level following the pre-registered analysis, Abadie et al. (2017)[71] and Gomila (2020)[72]) to address, we find results mirror accuracy by modality where transcripts have the lowest confidence score of 58% ($z(16004) = 69.3$, $\beta = 58$, $p < 0.001$, 95% CI = [56, 59]), silent videos are 7 percentage points higher ($z(16004) = 5.5$, $\beta = 7$, $p < 0.001$, 95% CI = [4, 9]), silent videos with subtitles are 9 points higher ($z(16004) = 7.8$, $\beta = 9$, $p < 0.001$, 95% CI = [7, 11]), audio (with and without subtitles) is 19 points higher ($z(16004) = 18.5$, $\beta = 19$, $p < 0.001$, 95% CI = [17, 22]), and video with audio (with and without subtitles) is 25 points higher ($z(16004) = 23.0$, $\beta = 25$, $p < 0.001$, 95% CI = [23, 27]). In columns 2 and 3 of Supplementary Table 1, we present results for real and fabricated speeches by themselves, which shows that additional media modalities help participants identify fabrications as fabrications even more than additional media modalities help participants identify real speeches as not fabricated.

As a secondary analysis using OLS, we find the participants' accuracy increased by 2.1 percentage points ($z(16002) = 4.6$, $\beta = 2.1$, $p < 0.001$, 95% CI = [1.2, 3.0]) for each of the three Cognitive Reflection Test (CRT)[73] questions they answered correctly. In addition, we find the participants' accuracy increases by 0.12 percentage points ($z(16002) = 3.9$, $\beta = 0.12$, $p < 0.001$, 95% CI = [0.06, 0.18]) for every stimulus seen.

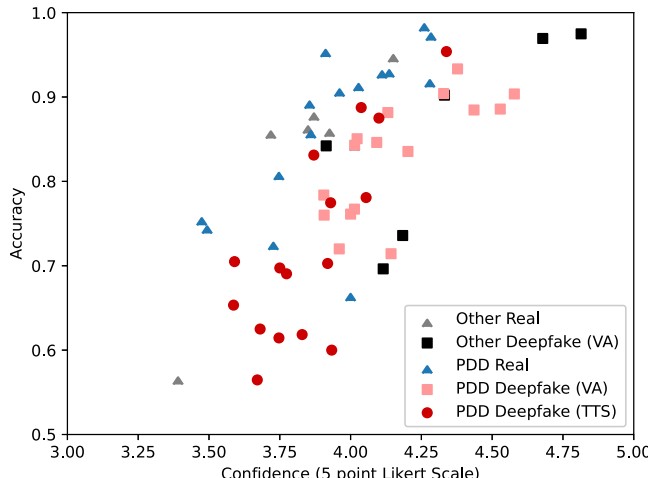

**Fig. 3 | Average accuracy and confidence for all video stimuli in experiment 2.** Scatter plot showing participants' mean accuracy and confidence (*N* = 5964 observations) across the 60 videos in Experiment 2. PDD indicates videos from the enhanced Presidential Deepfake Dataset, and Other indicates videos are drawn from the same sample used in Barari et al. 2021. VA indicates voice actor deepfakes and TTS indicates text-to-speech deepfakes.

## Experiment 1b (41,313 participants, not preregistered)

Experiment 1b presents a robustness check of Experiment 1a and is identical to Experiment 1a, except Experiment 1b has two orders of magnitude more participants and is not pre-registered. The results of Experiment 1b directionally corroborate results from Experiment 1a. Specifically, accuracy by modality from lowest to highest starts with transcripts at 43% accuracy followed by silent videos (with and without subtitles) at 65% accuracy, audio (with and without subtitles) at 74% accuracy, video with audio and subtitles at 82% accuracy, and video with audio but no subtitles at 83% accuracy.

## Experiment 2 (302 participants, preregistered)

In order to evaluate the generalizability of the results from Experiment 1, we designed and curated an enhanced and extended set of stimuli. In addition, we adapted Experiment 2 such that participants are not informed about the base rate of deepfakes, participants are not informed of whether stimuli are real or fabricated until the end of the experiment when we debrief participants, and we slightly adjust the experimental interface (see Methods section for details). These changes in Experiment 2 allow us to address the following question: How do manipulation methodologies and context influence participants' ability to discern real and fabricated political speeches on an enhanced and extended set of stimuli?

In Experiment 2, we show participants 20 videos randomly sampled from the 60 enhanced PDD videos and videos used in Barari et al. 2021 and ask participants whether they think the speech is fabricated, and how confident they are in their judgment. Each participant sees 4 real videos from the PDD, 4 voice actor deepfakes, 4 text-to-speech deepfakes, 4 real videos used in Barari et al. 2021, and 4 voice actor deepfakes used in Barari et al. 2021.

We find participants are less accurate on text-to-speech deepfakes than voice actor deepfakes, but we do not find statistically significant differences between participants' accuracy at identifying real videos as real and voice actor deepfakes as fabricated. In Supplementary Table 2, we present the pre-registered OLS regressions on accuracy, which is a binary variable defined as 1 if participants accurately identify the stimulus and 0 otherwise. Specifically, we do not find a statistically significant difference between accuracy on Barari et al. 2021 deepfakes and enhanced PDD voice actor deepfakes (z(5959) = −0.41, β = −0.02, p = 0.685, 95% CI = [−0.1.0.1]), real PDD videos (z(5959) = 0.25, β = 0.01,

p = 0.804, 95% CI = [−0.1.0.1]), or real videos used in Barari et al. 2021 (z(5959) = −0.30, β = −0.02, p = 0.763, 95% CI = [− 0.1.0.1]). However, we find that participants' accuracy on text-to-speech deepfakes is 13 percentage points lower than their accuracy on the deepfakes used in Barari et al. 2021 (z(5959) = −2.5, β = −0.13, p = 0.013, 95% CI= [−0.23, −0.03]).

In a series of 16 pre-registered two-sided *t* tests comparing PDD voice actor deepfakes with their text-to-speech counterparts, we find the accuracy on 5 out of 16 deepfakes are statistically significant and lower on text-to-speech videos than voice actor videos when controlling the false discovery rate using the Benjamini-Hochberg procedure[74]. Supplementary Table 3 presents the accuracy rates across each of the 16 enhanced deepfakes from the two audio sources alongside *p*-values from the two-sided *t* tests and the number of observations for each video.

Figure 3 presents the distribution of accuracy and confidence (as reported on a 5-point Likert scale) for each of the 60 videos in Experiment 2. In particular, this scatterplot demonstrates relatively high variance in accuracy across contexts; based on an OLS regression, accuracy and confidence are positively correlated (z(5964) = 7.5, β = 0.65, p = 0.001, 95% CI = [0.48, 0.81]). The real video with the lowest accuracy is a hot-mic of Obama speaking with Dimitri Medvedev[75], and the deepfake with the lowest accuracy is Donald Trump (as voiced by a text-to-speech algorithm) speaking about his phenomenal respect for women. The deepfakes with the highest accuracy (97%) are the two deepfakes of Donald Trump used in Barari et al. 2021.

We present the pre-registered secondary analysis using OLS in Supplementary Table 4 where we examine correct confidence (a weighted measure of accuracy defined as 1-(5-confidence)/5 if correct and -(confidence)/5 if incorrect), response time, and a binary variable for toggling the play/pause button. We find the results on correct confidence corroborate the main analysis on the accuracy, and we do not find any statistically significant effects of stimuli context on response time or toggling the play/pause button.

Based on OLS, we do not find statistically significant differences in the participants' accuracy changes over the course of the number of videos watched (z(5958) = 0.22, β = 0.0002, p = 0.828, 95% CI = [−0.001, 0.002]).

## Experiment 3 (1006 participants, preregistered)

In order to further evaluate the generalizability of the results in Experiment 1 and how base rates of misinformation may influence these results, we conduct Experiment 3 following the protocol in Experiment 2 where we show participants 20 political speeches randomly sampled from the 32 PDD political speeches and ask participants whether they think the speech is fabricated and how confident they are in their judgment. The deepfakes are all enhanced text-to-speech deepfakes from the PDD. We randomized participants to high and low base rate conditions where participants either see 16 or 4 fabricated speeches, respectively. Just like Experiment 2, we do not inform participants of the base rate of fabricated speeches and we do not inform participants of whether stimuli are real or fabricated until the end of the experiment when we debrief participants.

We do not find that the base rate of deepfakes has statistically significant effects on participants' overall accuracy, and we continue to find accuracy increases as participants have access to additional communication modalities. In Supplementary Table 5, we present the pre-registered OLS regressions on accuracy, which is a binary variable defined as 1 if participants accurately identify the stimulus and 0 otherwise. In columns 2 and 3 of Supplementary Table 5, we find the high base rate of fakes leads to a 7.2 percentage point higher accuracy on real stimuli (z(9702) = 4.6, β = 0.07, p < 0.001, 95% CI = [0.04, 0.10]) and 5.8 percentage point lower accuracy on fabricated stimuli (z(10100) = −3.6, β = 0.06, p < 0.001, 95% CI = [−0.09, −0.03]). When considering interactions in columns 4−6, we do not find statistically

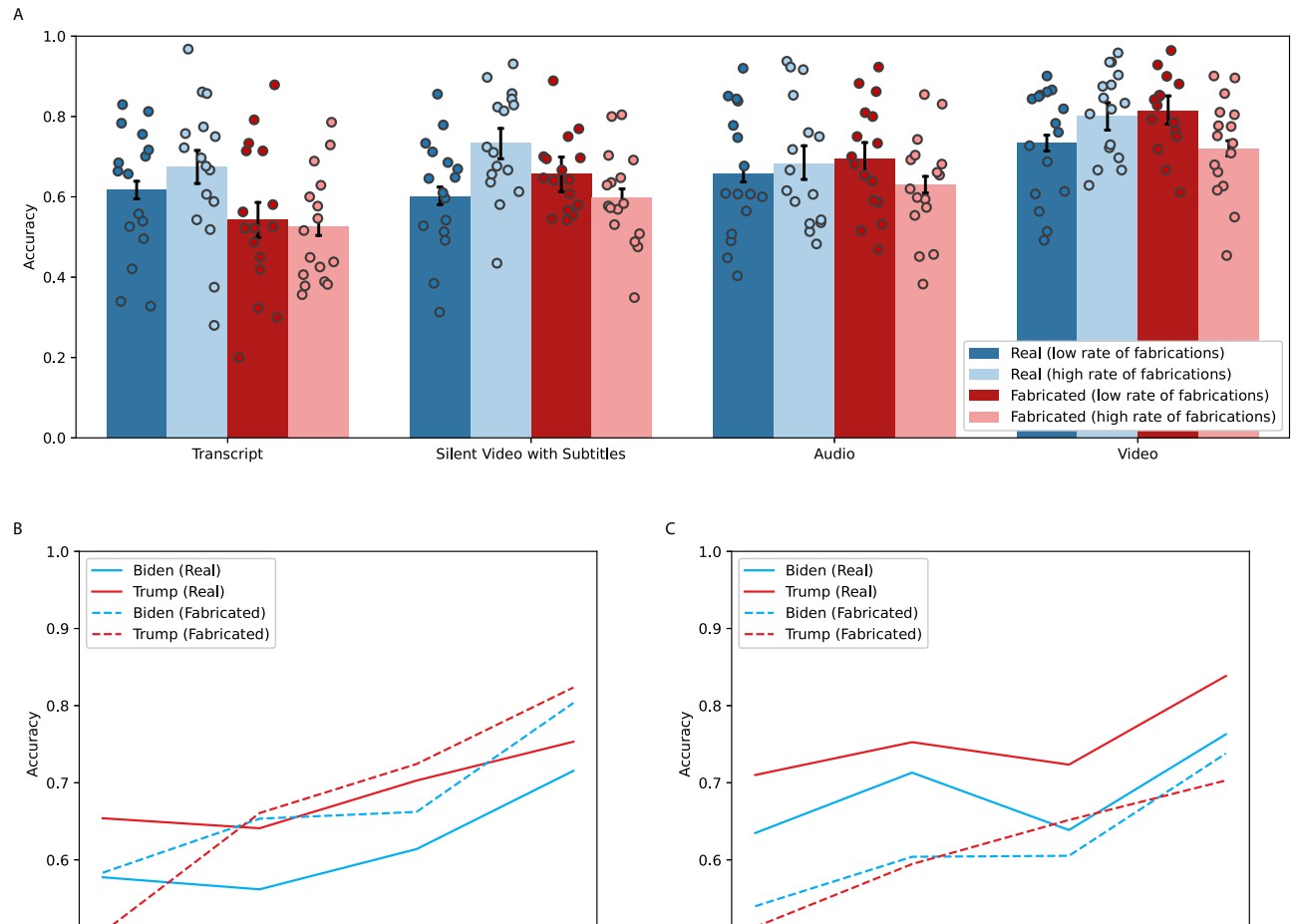

**Fig. 4 | Accuracy distinguishing real and fabricated speeches across media modalities and base rates in experiment 3. A** Mean accuracy across all permutations of text, audio, and video and high and low-base rate conditions in Experiment 3 ($N = 19,812$ observations). The error bars represent 95% confidence intervals. Dot plots represent the mean accuracy for each video stimulus. **B** Low base rate condition in Experiment 3: Mean accuracy across the four modalities ($N = 9572$ observations). **C** High base rate condition in Experiment 3: Mean accuracy across the four modalities ($N = 10,240$ observations).

significant differences between participants' overall accuracy on silent videos with subtitles and transcripts ($z(19804) = 0.6$, $\beta = 0.01$, $p = 0.532$, 95% CI = [−0.02, 0.04]), but we find silent videos with subtitles increase participants' accuracy on fabricated speeches by 11.4 percentage points ($z(10097) = 2.7$, $\beta = 0.11$, $p = 0.006$, 95% CI = [0.03, 0.20]). We find audio increases participants' overall accuracy by 6.3 percentage points ($z(19804) = 3.0$, $\beta = 0.06$, $p = 0.003$, 95% CI = [0.02, 0.11]) and specifically increases accuracy on real speeches by 4.1 percentage points ($z(9699) = 1.8$, $\beta = 0.04$, $p = 0.072$, 95% CI = [−0.004, 0.09]) and fabricated speeches by 15.2 percentage points ($z(10097) = 3.2$, $\beta = 0.15$, $p = 0.002$, 95% CI = [0.06, 0.25]). We find the largest impact on accuracy from video with audio which increases overall accuracy by 14.8 percentage points ($z(19804) = 3.0$, $\beta = 0.15$, $p < 0.001$, 95% CI = [0.10, 0.20]) and specifically increases accuracy on real speeches by 11.7 percentage points ($z(9699) = 4.5$, $\beta = 0.12$, $p < 0.001$, 95% CI = [0.18, 0.37]) and fabricated speeches by 27.2 percentage points ($z(10097) = 35.5$, $\beta = 0.27$, $p < 0.001$, 95% CI = [0.10, 0.20]). Figure 4 shows accuracy across base rates and modalities for real and fabricated stimuli for Experiment 3.

In order to evaluate the robustness of media effects on individual speeches, we conduct 6 families of comparisons of 32 pre-registered two-sided $t$ tests comparing accuracy in one modality to accuracy in another modality. Supplementary Tables 6–11 present the accuracy rates across each of the 32 political speeches alongside p-values from the two-sided $t$ tests between modalities and the number of

observations for each political speech in each modality. We find the accuracy on 6, 9, and 21 out of 32 political speeches are statistically significant and lower on transcripts than silent videos, audio, and video with audio, respectively, when controlling the false discovery rate using the Benjamini-Hochberg procedure[74]. Likewise, we find the accuracy on 4 and 19 out of 32 political speeches is statistically significant and lower on silent video than audio and video with audio. Finally, the accuracy on 12 out of 32 political speeches is statistically significant and lower on audio than video with audio.

We present the pre-registered secondary analysis using OLS in Supplementary Table 12 where we examine correct confidence, response time, and a binary variable for toggling the play/pause button. We find the results on correct confidence corroborate the main analysis on accuracy. We do not find any statistically significant effects of the base rate condition on response time or toggling the play/pause button. We find silent video takes participants an additional 7.0 s when viewing silent video relative to audio ($z(14323) = 3.8$, $\beta = 7.0$, $p < 0.001$, 95% CI = [3.4, 10.6]), and we find silent video and video with audio leads participants to toggle the play and pause 15.1 and 3.0 percentage points ($z(14323) = 12.1$, $\beta = .15$, $p < 0.001$, 95% CI = [0.13, 0.18] and $z(14323) = 3.2$, $\beta = .03$, $p = 0.002$, 95% CI = [0.01, 0.05]) more than audio, respectively.

Based on OLS, we do not find statistically significant differences in the participants' accuracy over the course of the number of stimuli seen in either condition ($z(19802) = −1.2$, $\beta = −0.001$, $p = 0.231$, 95%

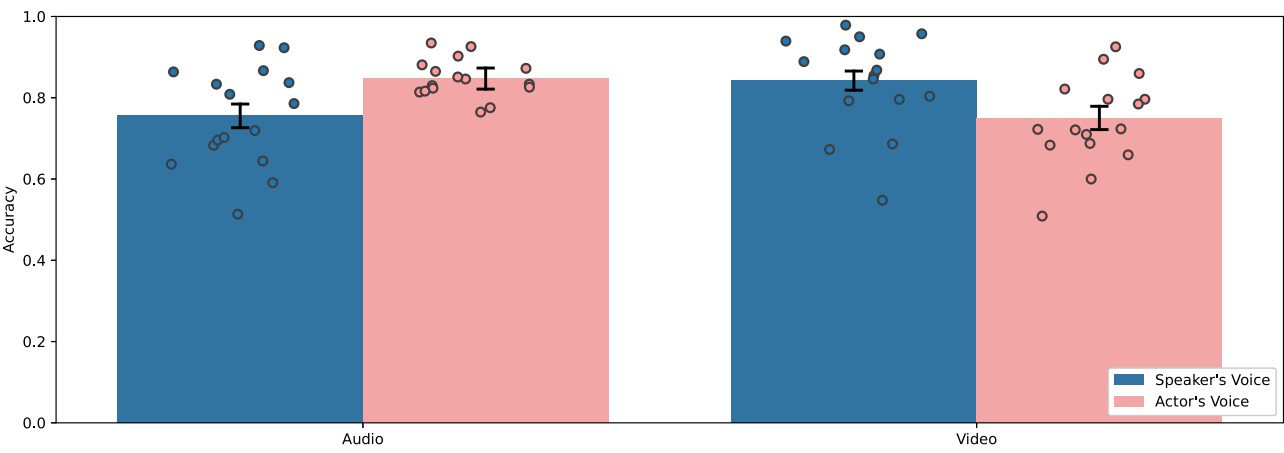

**Fig. 5 | Accuracy distinguishing speakers' and actors' voices across video stimuli in experiment 4.** Mean accuracy across all audio and video in Experiment 4 ($N = 3215$ observations). The error bars represent 95% confidence intervals. Dot plots represent the mean accuracy for each video stimulus.

CI = [−0.003, 0.001]) or high base rate condition by itself ($z(19802) = 0.2$, $\beta = 0.0002$, $p = 0.881$, 95% CI = [−0.002, 0.003]).

## Experiment 4 (206 participants, preregistered)

In order to further identify the role of manipulated audio in participants' ability to distinguish between real and fabricated content across the previous experiments, we designed Experiment 4 to address the following question: How does media modality influence participants ability to distinguish between a well-known speaker's real voice and an actor's voice? In Experiment 4, we show participants 16 real PDD political speeches and ask participants whether they think the voice is the speaker's or a voice actor, and how confident they are in their judgment. By focusing on only real videos, we examine the role of perfectly realistic visual information to influence accuracy. Just like Experiments 2 and 3, we do not inform participants of the base rate of voice actor audio and we do not inform participants of whether stimuli are the actual speakers' or voice actor's voice until the end of the experiment when we debrief participants.

We find that participants are more accurate at identifying voice actors' audio than real speakers' audio, more accurate on real speakers' video than real speakers' video, but less accurate on voice actor video with audio than voice actor audio by itself. In Fig. 5 and Supplementary Table 13, we present the pre-registered OLS regression analysis on accuracy. Specifically, participants are 75.6% accurate on audio by the actual speakers and we find voice actor audio increases accuracy by 9.2 percentage points ($z(3211) = 2.7$, $\beta = 0.09$, $p = 0.006$, 95% CI = [0.03, 0.16]), video with audio increases accuracy by 8.8 percentage points ($z(3211) = 2.9$, $\beta = -0.09$, $p = 0.004$, 95% CI = [0.03, 0.15]), but the combination of video with audio by voice actors lowers accuracy by 18.7 ($z(3211) = -4.3$, $\beta = -0.19$, $p < 0.001$, 95% CI = [−0.27, −0.10]) percentage points such that it is the same level of accuracy as participants obtain on audio (without video) by the actual speakers. Despite identical base rates of voice actor audio in the video and audio conditions and similar overall accuracy rates of 80% in both video and audio conditions, participants were biased to identifying audio stimuli as an actor's voice in 55% of observations and identifying video stimuli as an actor's voice in 45% of observations. This suggests the voice actor's audio consistently matched with the real video footage, and audio with a video increased participants' beliefs that all audio (both real and fake) was authentic relative to when participants were listening to the audio only.

We present the pre-registered OLS secondary analysis in Supplementary Table 15 where we examine correct confidence, response time, and a binary variable for toggling the play/pause button. We find the results on correct confidence corroborate the main analysis on

accuracy. We do not find effects on toggling the play/pause button, but we find that voice actor audio leads participants to take an additional 1.5 seconds ($z(3211) = -3.8$, $\beta = -1.5$, $p < 0.001$, 95% CI = [−2.3, −0.7]) beyond what they take for audio by the real speakers.

For each additional stimulus seen, we find via OLS that the participants' accuracy increases by 0.3 percentage points ($z(3210) = 2.6$, $\beta = 0.003$, $p = 0.011$, 95% CI = [0.001, 0.005]); participants slightly improve based on seeing and hearing real and fabricated stimuli.

## Experiment 5 (200 participants, preregistered)

In contrast to Experiments 1 through 4 where participants are explicitly asked to consider the veracity of the stimulus, Experiment 5 is designed not to alert participants to the dependent variable of interest. Instead in Experiment 5, participants are asked, "What comes to mind after watching the following video/listening to the following audio/reading the following quote?" Instead of the custom website at detectfakes.media.mit.edu used in the previous experiments, Experiment 5 is hosted on Qualtrics and we changed the initial instructions as follows: "This is an MIT research project. You will be shown quotes, audio, and video files that one might expect to see on social or digital media. You will be requested to share your thoughts and opinions after reading, listening, or viewing each of these media files." Just like Experiments 2 through 4, we do not inform participants of the base rate of voice actor audio and we do not inform participants of whether stimuli are real or fabricated until the end of the experiment when we debrief participants.

In Experiment 5, each participant sees a forced choice attention check, the deepfake attention check, and 10 randomly sampled stimuli from Experiment 3 (32 real and fabricated speeches from the PDD where the 16 deepfakes are the text-to-speech deepfakes). In order to evaluate the effect of recent exposure to a deepfake, we randomly assign participants to see the deepfake attention check video at the start or end of the experiment. Additional details about the design of Experiment 5 and how free text responses are annotated into a binary variable for suspicion of fabrication are shared in the Methods section.

Similar to Experiment 1, we find that participants' accurate suspicion of a fabrication increases as participants have access to additional communication modalities. Further, we find that the false positive rate (inaccurate suspicion of fabrication) is not associated with communication modalities. Supplementary Table 17 presents pre-registered OLS regression results for Experiment 5 and column 2 of Supplementary Table 17 shows that relative to text transcripts, silent video increases the true positive rate by 10.3 percentage points ($z(1013) = 3.2$, $\beta = 0.10$, $p = 0.001$, 95% CI = [0.04, 0.17]), audio increases the true positive rate by 6.3 percentage points ($z(1013) = 2.3$, $\beta = 0.06$,

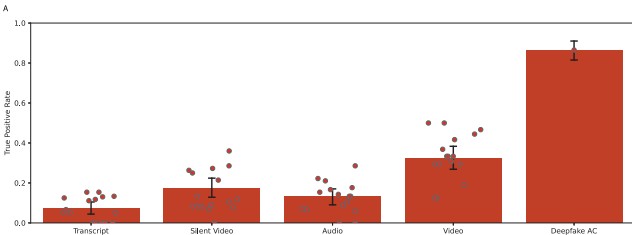
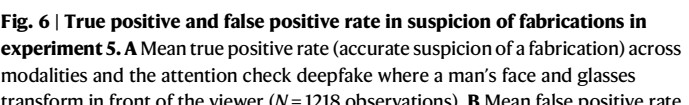

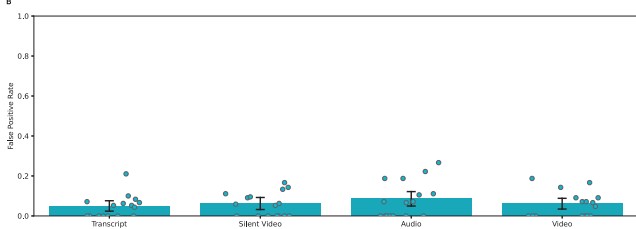

**Fig. 6 | True positive and false positive rate in suspicion of fabrications in experiment 5. A** Mean true positive rate (accurate suspicion of a fabrication) across modalities and the attention check deepfake where a man's face and glasses transform in front of the viewer (*N* = 1218 observations). **B** Mean false positive rate

(inaccurate suspicion of a fabrication) across modalities (*N* = 982 observations). The error bars represent 95% confidence intervals. Dot plots represent each video stimulus.

*p* = 0.022, 95% CI = [0.01, 0.12]), video increases the true positive rate by 25 percentage points (z(1013) = 7.3, *β* = 0.25, *p* < 0.001, 95% CI = [0.19, 0.32]), and priming (assignment to the attention check deepfake displayed as the first stimulus as opposed to the last) is not statistically significant at the *p* < 0.05 threshold but is associated with an increase in the true positive rate by 7.4 percentage points (z(1013) = 1.9, *β* = 0.07, *p* = 0.057, 95% CI = [−0.002, 0.15]). In contrast, the same regression on the false positive rate does not have significant values on any of the modality conditions but priming increases the false positive rate by 6.3 percentage points (z(977) = 3.3, *β* = 0.06, *p* = 0.001, 95% CI = [0.03, 0.10]). In Fig. 6, we present the true positive and false positive rates for suspicion of fabrications in the experiment. In 32.4% of observations of deepfake videos with audio, participants' responses were judged to be suspicions of fabrications whereas the rate of suspicions was only 6 percentage points in real videos with audio. Audio, by itself, is much less likely to reveal suspicions: 13.0% of observations of text-to-speech audio are suspected whereas 8.6% of observations of real audio are suspected. Finally, transcripts are the least likely modality to reveal suspicions; 7.3% of observations of fabricated transcripts show suspicions compared to 4.8% of observations of real transcripts.

We note that the vast majority of participants' responses (86.5%) to the obvious, attention check deepfake indicate participants suspected the video included a fabrication. In addition, we note we did not pre-register heterogeneous analyses based on age but we find evidence that accurate suspicion of a deepfake is correlated with age. In particular, the true positive rate across participants' ages ranges from 31% for 32 people in their 20s, 43% for 54 people in their 30 s, 31% for 48 people in their 40s, 25% for 24 people in their 50s, and 20% for people in their 60s and beyond.

## Discussion

This paper provides evidence, via multiple pre-registered randomized experiments with 2215 participants that visual and auditory communication modalities increase people's ability to distinguish authentic political speeches from fabricated political speeches. In the context of authentic speeches, we provide corroborating evidence for the conventional wisdom around the seeing is believing narrative (the realism heuristic that suggests people will tend to trust video over text[31] and results from Wittenberg et al. 2021 showing people "are more likely to believe an event occurred when it is presented in video versus textual form"[30]); people are significantly more accurate at identifying authentic speeches as authentic when the speeches include audio and visual modalities as opposed to only text (although note Wittenberg et al. 2021 finds minimal effects of video on persuasiveness). However, with respect to fabricated content, the results from Experiments 1–3 and 5 add considerable nuance to the seeing is believing narrative: people are significantly more accurate at identifying fabricated speeches as fabricated when the speeches include audio and visual modalities as opposed to only text. In other words, we find participants are significantly more accurate at distinguishing between authentic and

fabricated political videos than transcripts. Moreover, Experiment 5 demonstrates that this continues to be the case even when people are not directly asked about the authenticity of a speech or are primed to consider accuracy.

These results are based on an experiment with a stimuli set that is much larger than most stimuli sets for the psychology of media effects research[38] and deepfake detection[33,35,36], but it is important to add a caveat that we focused on a single context, political speeches, and a combination of algorithms, the deepfake lip-syncing wav2lip algorithm and the DeepFaceLab library, which are very effective at manipulating a person who is facing forward and already speaking into a convincing deepfake video. While we present evidence that adds considerable nuance to the media effects literature on communication modalities, future work may consider additional nuances by exploring heterogeneity based on other kinds of deepfake manipulations like face swapping and head puppetry[76], contexts that require more sophistication to produce a convincing deepfake (e.g., where a person is moving, turning their head, and interacting with other people), who is being manipulated[77], and contexts immediately relevant to current events (for example in March 2023, fake arrest images of Donald Trump were released on social media leading up to his indictments by the Manhattan District Attorney's office and the United States Department of Justice[78]).

The results from these experiments cannot simply be explained by the deepfake manipulations being too obvious or unrealistic. Figure 1 illustrates accuracy across the political speeches displayed as video with audio and reveals that while participants are relatively highly accurate at identifying voice actor deepfakes, participants only identify text-to-speech deepfakes in 72% of observations, which is closer to random guessing than a perfect score. Moreover, human discernment is influenced by a number of factors, and we find 10 text-to-speech PDD deepfakes, 2 voice actor PDD deepfakes, and 2 deepfake videos used by Barari et al. 2021 are accurately identified in less than 75% of observations. The participants' low accuracy offers evidence that visual artifacts and inconsistencies created by the lip-syncing deepfake manipulations are not readily apparent to most people, and as such, these videos represent reasonable stimuli set for examining how well people can distinguish real from deepfake videos and how communication modalities, audio sources, and base rates of misinformation influence discernment.

People distinguish authentic from fabricated videos based on perceptual cues from video and audio and considerations about the content (e.g., the degree to which what is said matches participants' expectations of what the speaker would say, which is known as the expectancy violation heuristic[79]). With the message content alone, participants are only slightly better than random guessing at 57% accuracy in Experiment 1a and 58% in Experiment 3. With perceptual information from video and the message content via subtitles, participants are slightly more accurate at 66% accuracy in Experiment 1a and 62% accuracy in Experiment 3. With information from audio only,

participants are more accurate at 80.5% in Experiment 1a and 65% in Experiment 3. Finally, with information from both video and audio, participants are even more accurate at 82% accuracy in Experiment 1a and 74% accuracy in Experiment 3. Our finding that participants are more accurate at distinguishing between real and fabricated voice actor audio than silent video with subtitles aligns with the social psychology literature demonstrating people tend to rely on auditory information more than visual information for both discerning sincerity[80] and ascribing authorship of a script to a human (as opposed to a computer)[81]. Another factor that could be supporting improved detection of deepfakes in the audiovisual regime is the beneficial effect of multisensory integration. Complementary audio-visual information has been shown to improve accuracy on perceptual decision-making tasks compared with visual information only, by amplifying post-sensory decision evidence[82]. However, the low accuracy of participants in distinguishing the speaker's audio from text-to-speech audio trained on the speaker suggests social cues oriented towards speech in digital interfaces will need to adapt to fabricated audio that is nearly indistinguishable from real audio. Overall, the experiment's results show that as participants have access to more information via audio and video, they are better able to distinguish whether a political speech has been fabricated.

Political deepfakes are most dangerous when people are least expecting information to be manipulated, and these experiments examine the influence of the base rate of misinformation on participant discernment. In Experiment 1, 50% of the content is fake, and we explicitly inform participants of this base rate. In Experiment 3, the base rate of misinformation is randomized to be 20% or 80% of stimuli and we do not inform participants of this base rate. We find the high base rate of fakes compared to the low base rate leads participants to a 7.2 percentage point higher accuracy on real stimuli and a 5.8 percentage point lower accuracy on fabricated stimuli, which are both statistically significant. In other words, participants responded that both real and fabricated stimuli are fake less often in the condition with the high base rate of misinformation than the low base rate. One explanation for this difference may be that people generally do not expect a very high base rate of fakes, which may lead people to respond that a stimulus is fake less often than they would in a more balanced setting. While false news is relatively rare in today's media ecosystem and is approximated to make up less than a fraction of a percent[83,84] of news, the capacity for generative AI to create misinformation is expanding[85], and future base rates of misinformation may be higher.

The political danger of fabricated videos may not be the average algorithmically produced deepfake but rather a single, highly polished, and extremely convincing video. For example, hyper-realistic deepfakes such as the Tom Cruise deepfakes on Tiktok under the username @deeptomcruise are produced by visual effects artists using artificial intelligence algorithms but also using traditional video editing software and highly trained look-alike actors. While these hyper-realistic deepfakes may still contain manipulation artifacts (e.g., unattached earlobes that do not match Tom Cruise's attached earlobes[86]), future work on the psychology of multimedia misinformation may consider hyper-realistic videos produced by visual effects studios in addition to algorithmically manipulated videos. Experiment 4 offers insights on the future influence of hyperrealistic videos by demonstrating that perfectly realistic video (the real videos paired with voice actor audio) leads people to believe audio is more likely to be authentic than when listening to audio by itself.

These experiments are useful for studying how people discern multimedia information when attending to questions of accuracy, but they are less useful in understanding how people will share misinformation they consume on social media. People are generally highly accurate in discerning the veracity of news headlines yet share false news headlines because their attention is not focused on accuracy[87]. In

fact, Epstein et al. 2023 show that simply considering whether to share news on social media decreases people's accuracy at truth discernment[88]. Similarly, our findings in experiment 5 reveal that priming participants with a video showcasing visual effects manipulations leads people to express slightly more suspicions in free responses than participants who have not yet seen such a showcase video. These results support recent research showing that educational material on common misinformation techniques can improve people's ability to discern trustworthiness from untrustworthy videos[89].

On social media, video-based misinformation will often be designed to incorporate characteristics (e.g., fear, disgust, surprise, novelty) that divert people's focus from accuracy and make content go viral[90–93]. Given that multimedia misinformation may be both easier to discern and more frequently shared on social media than text-based media, more research needs to be done to understand how people allocate attention while browsing the Internet[94]. Our findings in Experiment 5 (which presents an environment much more similar to social media than the previous experiments) suggest that many people pay attention to the question of authenticity and remark on suspicions even without being asked about a stimulus' authenticity.

It is important to keep in mind that discernment – how accurately people discern misinformation – is different than belief – how much people report they believe misinformation. It is possible (though quite peculiar) that someone could be highly accurate at discerning truth from falsehood while also tending to believe the fabricated content and not believe the true content. For example, research on false news headlines and articles finds that people are better at discerning news concordant with their political leanings than discordant news while also believing concordant news more often than discordant news[64].

The findings that videos of political speeches are easier to distinguish as authentic or fabricated than text transcripts highlight the need to re-introduce and explain the oft-forgotten second half of the seeing is believing adage. In 1732, the old English adage appears as: "Seeing is believing but feeling is the truth."[95] Here, the feeling does not refer to emotion but rather the direct experience. Since the advent of photography, people across society have generally understood that what we see in a photograph is not always the truth and further assessment is often necessary[96–98].

In this paper, we examined a bounded question – how well can ordinary people discern (and how often do they suspect) whether or not a short soundbite of a political speech by a well-known politician in text, audio, or video has been fabricated – and we find that more information via communication modalities – text transcripts vs. silent, subtitled video vs. video with audio – enables people to more accurately discern fabricated and real political speeches. These results are particularly relevant for the design of content moderation systems for flagging misinformation on social media. Future research should consider how explanations that indicate which component part of a video appears to be fabricated influence believability and sharing of fake content. These explanations could allow people to appropriately allocate attention to the content[99] or perceptual cues (e.g., low-level pixel features, high-level semantic features, and biometric-based features[100]) when trying to assess the content's authenticity.

Finally, these findings offer insights into political communication and communication theory more generally; there is more to how humans form beliefs than the seeing is believing narrative would suggest because people can pay attention and seek out inconsistencies in both what is said and how something is said.

## Ethical and societal impact

In this research, we created political speech deepfakes in order to experimentally evaluate what influences people's ability to distinguish between real political speeches and digitally fabricated political speeches. Given the sensitive nature of political speeches' potential to persuade people and the stimuli materials featuring some of the

leading candidates for the 2024 United States presidential election at the time of publication, we reflect on the positive potential outcomes and applications of this work as well as potential negative ones or unintentional misuse.

This work offers experimental evidence from a large experiment with 2215 participants and 44 political speeches for how realistic deepfakes are: people are significantly better than random guessing but far from perfect at distinguishing between deepfake political speeches and real speeches. Moreover, we find that many people pay attention to the question of a political speech's authenticity and remark on suspicions even without being directly asked about authenticity. These results stand in contrast to what people might expect of deepfake photorealism from seeing the most popular deepfakes online, such as the Tom Cruise deepfakes seen by tens of millions of people. As such, these results help inform an emerging branch of media literacy, generative AI literacy[101,102], by offering both an opportunity for people to calibrate both their own and other's ability to detect deepfakes. In addition, these results offer an important nuance to scholarship on the persuasiveness of video versus text by revealing that the conventional wisdom that seeing is simply believing and people will fall for fake news more often when the same version of a story is presented as a video versus text does not hold up to empirical scrutiny. Finally, this research offers insights into how base rates of misinformation influence discernment, how the realism of voice actor audio compares to text-to-speech algorithms, how question framings influence discernment rates, and how discernment of deepfakes varies depending on the specifics of the content.

The insights from this research cannot be fully disentangled with potential negative outcomes related to the risk of sharing deepfake stimuli outside the context of the research experiments, which could result in spreading misinformation. In order to mitigate the risk of sharing outside the context of the research experiment and misinforming people, we implemented the following mitigation strategies: First, we avoided extremely sensational and inflammatory content when generating political speeches that might provoke moral outrage and increase the likelihood that these videos would be shared on social media[93]. Second, we posted the transcripts of all fake videos in the appendix of the publicly available The Presidential Deepfakes Dataset[103] in which the videos were first described such that anyone searching for the fake speeches online would immediately find them marked as fake. Third, we collaborated with TruePic to implement the C2PA protocol to cryptographically bind the videos with metadata, marking these videos as AI-generated. This metadata is attached to the original deepfakes, which allows a fact-checker to quickly identify the video as fake. Fourth, we have not shared methodological details for producing deepfakes in this manuscript per request of the editorial team to reduce potential misuse that could arise with detailed know-how for producing deepfakes. These methodological details appeared in earlier versions of the manuscript, and this information is now available upon request. Fifth, we make the enhanced deepfake videos available only upon request like other related digital forensics research projects have done[5].

## Methods
### Consent and ethics
This research complied with all relevant ethical regulations and the Massachusetts Institute of Technology's Committee on the Use of Humans as Experimental Subjects approved this study as Exempt Category 3 – Benign Behavioral Intervention. This study's exemption identification numbers are E-3105 and E-3354 for Experiment 1, E-4735 for Experiments 2, 3, and 4, and E-5493 for Experiment 5. For experiments 1 through 4, all participants were presented with an informed consent statement: "Detect Fakes is an MIT research project. All guesses will be collected for research purposes. All data for

research are collected anonymously. For questions, please contact detectfakes@mit.edu. If you are under 18 years old, you need consent from your parents to use Detect Fakes."

For participants in experiment 5, the informed consent and instructions statement was presented as follows: "This is an MIT research project. You will be shown quotes, audio, and video files that one might expect to see on social or digital media. You will be requested to share your thoughts and opinions after reading, listening, or viewing each of these media files. All response data is collected anonymously for research purposes... Participation is voluntary, and you may only participate if you are 18 years of age or older. For questions, please contact arunas@mit.edu."

Before beginning any of the experiments, all participants from Prolific were also provided a research statement, "The findings of this study are being used to shape science. It is very important that you honestly follow the instructions requested of you on this task, which should take a total of 15 minutes. Check the box below based on your promise:" with two options, "I promise to do the tasks with honesty and integrity, trying to do them uninterrupted with focus for the next 15 minutes." or "I cannot promise this at this time." Participants who responded that they could not do this at this time were re-directed to the end of the experiment.

In Experiment 1, we immediately debriefed participants on which political speeches are real and which are fabricated after each video is seen. In Experiments 2 through 5, we debrief participants on which political speeches are real and which are fabricated at the end of the experiment. In order to limit the potential for these deepfakes to be taken out of their research context, we created a public website showing the deepfakes signed using the C2PA protocol to indicate these videos are partially AI-generated. If deepfakes were taken out of context, people can reference these signed deepfakes to identify them as fabrications designed for research.

For participants in Experiment 1a recruited from Prolific, we compensated participants at a rate of $9.78 an hour and provided bonus payments of $5 to the top 1% of participants in terms of accuracy. In Experiment 1b, we did not compensate participants financially because they arrived at the website via organic links on the Internet. In Experiments 2 through 5, all participants are recruited from Prolific and compensated at a rate of $12.00 an hour.

### Digital experiment interface
In experiments 1 through 4, we hosted multimedia stimuli – transcripts, audio, and video of authentic and fabricated political speeches – on a custom-designed website called Detect Fakes, which was hosted at https://detectfakes.media.mit.edu/. In these experiments, we asked participants to identify stimuli as fabricated and non-fabricated. In experiment 5, we used Qualtrics and asked participants "What comes to mind after watching the following video/listening to the following audio/reading the following quote?"

**Experiments 1a and 1b.** First, we collected informed consent, presented participants with instructions and an attention check, and then we showed participants a short political speech and asked "Did [Joseph Biden/Donald Trump] say that?" followed by "Please [read/listen/watch] this [transcript/audio clip/video] from [Joseph Biden/Donald Trump] and share how confident you are that it is fabricated. Remember, half the media snippets we show are real and half are fabricated." Participants were instructed to move a slider to report their confidence from 50% to 100% that a stimulus is fabricated (or 50% to 100% that a stimulus is not fabricated). After each response, we presented feedback to participants to inform them whether the stimulus was actually fabricated. Then, we presented participants with another stimulus (selected at random) and the process repeated until participants viewed all 32 stimuli or decided to leave the experiment.

**Experiments 2, 3 and 4**. Similar to Experiment 1, we first collected informed consent and presented participants with instructions and an attention check. Next, we showed participants a short political speech and asked "Fabricated or Authentic?" followed by "Please [read/listen/watch] this [transcript/audio clip/video] from [speaker] and share whether you think the speech is fabricated, and how confident you are in this judgment." In Experiment 4, we edited the follow-up to "Please watch and listen to this [audio clip/video] from [speaker] and share whether you think the voice is [speaker's] or a voice actor, and how confident you are in this judgment." Participants chose between two options – Real or Fake – and reported their confidence along a Likert scale from 1 = Not confident at all to 2 = Slightly confident, 3 = Somewhat confident 4 = Fairly confident to 5 = Completely confident. The experiment interface prevented participants from selecting the real or fake radio button until they had watched at least 15 seconds of the audio or video, prevented participants from selecting their confidence rating until they had selected a radio button, and prevented participants from submitting their response until they have selected a radio button and confidence rating.

**Experiment 5**. In Experiment 5, we began by collecting informed consent and presenting participants with instructions and a directed choice attention check. Instead of directly asking participants to identify whether the video is real or fake, Experiment 5 asks, "What comes to mind after watching the following video/listening to the following audio/reading the following quote?" Participants are presented with a text response box. In order to proceed to the next stimulus, the interface requires participants to write at least 50 characters and spend at least 30 s on each stimulus. Experiment 5 is hosted on Qualtrics (as opposed to our custom website described in the Methods section) to prevent participants from knowing the dependent variable of interest in this experiment.

In order to create the key dependent variables for this experiment, three authors of this article independently annotated 2200 participants' responses for whether participants expressed suspicion that the stimulus was fabricated or not. The dependent variable represents the majority agreement between the three annotators. Cohen's Kappa between annotators on PDD stimuli as a whole were 0.56, .59, and 0.85; the Cohen's Kappa between annotators is 0.35, 0.39, and 0.72 on text, 0.49, 0.51, and 0.79 on silent video, 0.58, 0.62, and 0.87 on audio, and 0.69, 0.70, and 0.90 on video with audio. Two annotators identified suspicion in 11% of observations and the third annotator identified suspicion in 22% of observations. Across the 2,000 observations, 9% are identified by all three as revealing suspicion, 3% are identified by two or three, 11% are identified by one, and 77% are identified by all three as not revealing suspicion. Examples of revealing suspicion by two authors but not a third include: "I'm not sure this is a real statement made by Trump" and "I'm not sure if Biden actually said this or not, but it sounds convincing enough." Likewise, examples that don't count as revealing suspicion because only one author annotated it as suspicion include: "If what he says were true then that would be a very beneficial thing." and "I'm not completely sure what Biden is trying to say here, I would want more information."

### Experiment stimuli
**Original presidential deepfakes dataset**. In Experiments 1a and 1b, the stimuli were drawn from the Presidential Deepfake Dataset (PDD)[68]. The PDD consists of 32 videos showing two United States presidents – Donald Trump and Joseph Biden – making political speeches. Half the videos are original videos that have not been altered or manipulated. The other half had been fabricated to make the politicians appear to say something that they have not said. The fabricated videos were produced by writing a fabricated script, recording professional voice actors reading the script, and applying a deepfake lip-syncing algorithm[17] to real videos of Joseph Biden and Donald Trump

to make it appear as if the politicians actually gave such a fabricated speech. The mean duration of the videos is 21 seconds and all videos are recorded at 30 frames per second. The PDD is balanced across three dimensions: (1) videos that had and had not been fabricated, (2) videos of Joseph Biden and Donald Trump, and (3) videos of the two politicians making concordant and discordant speeches with what the general public believes were the politicians' political views.

In order to validate the concordance and discordance of speeches, we conducted an independent survey where 84 participants who passed an attention check rated each of the 32 transcripts for how well the political speeches matched either politician's political views. Participants were instructed "For each statement, we want you to rank how closely the statement matches your understanding of President Joseph Biden or President Donald Trump's political views" and asked to provide a judgment on a 5-point Likert scale from "Strongly Disagree" (-2) to "Strongly Agree" (2) that "This statement matches President [Joseph Biden's/Donald Trump's] political viewpoint: [statement]." Participants' responses confirm that speeches designed to be concordant and discordant with the two politicians views were indeed concordant and discordant with the average participants' perception of the politicians' views. The $Z$-values of participants responses to concordant and discordant speeches are −0.25 and 0.21, respectively, and this difference is statistically significant with $p < 0.001$ based on a two-sided $t$ test.

In Experiment 1, we transformed each of the original videos from the PDD into 7 different forms of media: a transcript, an audio clip, a silent video, audio with subtitles, silent video with subtitles, video with audio, and video with audio and subtitles. As a result, there were 7 modality conditions, 32 unique speeches, and 224 unique stimuli. In the digital experiment, the transcript appears as HTML text and the six other forms of media content appear in a video player. The audio clip shows a black screen in the video player and the audio clip with subtitles shows a black screen with subtitles at the bottom.

**Enhanced presidential deepfakes dataset.** The enhanced PDD included the same real videos from the PDD and two sets of 16 enhanced deepfakes with voice actor audio and audio from a text-to-speech algorithm. The 16 enhanced deepfakes included source videos more amenable to deepfake lip-syncing manipulations than the initial source videos for the PDD deepfakes, visual touch-ups with the DeepFaceLab library, audio from both a voice actor and a text-to-speech algorithm trained on the speakers' voices, and additional audio engineering to add in background noise and subtle acoustic elements to make the audio appear real.

While methodological details for enhancing the PDD appeared in earlier versions of the manuscript, this information is now available upon request to reduce potential misuse that could arise from detailed know-how for producing deepfakes.

**Other stimuli.** We included 6 additional deepfakes and 6 additional real videos in Experiment 2, which are used in Barari et al. 2021. The deepfakes included two deepfakes of Bernie Sanders and Hilary Clinton from the Agarwal et al. 2019 dataset[5], which involve face swapping of the politicians' faces onto videos of Saturday Night Live skits by Larry David and Kate McKinnon. The four other deepfakes can be found on Youtube and showed Boris Johnson endorsing his opponent, Trump announcing we have eradicated AIDS, Trump announcing deepfakes would make it easy to make him say ridiculous things, and Obama announcing we are in an era in which our enemies can make it look like anyone is saying anything.

### Preregistration
Experiments 1a, 2, 3, 4, and 5 involved participants recruited from Prolific and were pre-registered on aspredicted.org at the following URLs: Experiment 1a (https://aspredicted.org/m45q9.pdf), Experiment

2, Experiment 3 (https://aspredicted.org/re82c.pdf), Experiment 4 (https://aspredicted.org/ke5r8.pdf), and Experiment 5 (https://aspredicted.org/pm9vw.pdf). In Experiment 1a, we pre-registered that we would "test the hypothesis that motivated reasoning plays an outsized role in video compared to other media" and we would evaluate this hypothesis by "interacting treatment effects with the party affiliation of the participant, speaker, and the content" but we do not provide this analysis because the interaction of treatment effects does not directly test the motivated reasoning hypothesis. Otherwise, there were no deviations from the pre-registrations.

Experiment 1b was not pre-registered and included 41,313 participants who discovered the experiment organically through search engines or news media.

## Participants

In Experiments 1a, 2, 3, 4, and 5, we recruited participants via the Prolific platform[104]. In each of these experiments, participants responded to a baseline survey, which consisted of questions on political preferences, experience with deepfakes, and trust in media and politics. Prolific provided basic demographic data including participants' self-reported sex, ethnicity, and age. In Experiment 1a but not Experiments 2–4, we included three questions from the Cognitive Reflection Test (CRT)[73]. In Experiment 1b, we collected data from participants who visited the experiment organically but did not collect demographic, political identity, or pre-experiment questions for these non-recruited participants.

In Experiments 1a, 2, 3, and 4, 95%, 83%, 84%, and 89% of participants provided responses to the complete set of stimuli, which results in 99.9%, 98.7%, 97.5%, and 97.6% of the expected data in each experiment. The missing data appears to be missing due to intermittent or slow network connection issues where participants could proceed without their data getting submitted to the server. In Experiment 1b where participants were not recruited and were not asked to complete a pre-specified number of responses, 15% of participants provided responses to the complete set of stimuli.

We exclude participants from participating in multiple experiments. However, 9 participants who participated in Experiment 2 also participated in Experiment 4. The results in Experiment 4 are robust to both including and excluding these 9 participants.

In these experiments, we do not find consistent differences based on sex, and we do not report sex-based analyses because we did not pre-register sex-based analyses nor do we have theoretical grounds for suspecting differences across sex.

**Experiment 1a – March 20 to April 8, 2021.** In Experiment 1a, we recruited 501 participants from the United States who successfully passed the attention check and provided 16,011 observations. In Experiment 1, the expected sample size of 500 participants providing responses to 32 stimuli each provides 16,000 observations split between 7 conditions, which further provides over 90% statistical power to detect differences between conditions of 5 percentage points. All following experiments keep this statistical power in mind. The demographic distribution of participants along sex and age is: 50% male, 49% female, and 1% unknown; 60% 18 to 35, 37% 36−64, and 3% over 64. We do not have data on participants' race or ethnicity. The sample of 509 recruited participants is balanced across political identities: 255 recruited participants self-report as Democrats, and the other 246 recruited participants self-report as Republicans. In response to a pre-experiment question on participants' experience with deepfakes, fewer than 1% of participants responded that they have created their own deepfakes, 73% of participants have seen a few to several examples of deepfakes, and 27% of participants have yet to see their first deepfake or don't know their experience with deepfakes. In terms of trust and confidence in media, 43% of participants report a fair amount or a great deal of trust and confidence in media and 57% of

participants report not very much or none at all. On the topic of following news on government and public affairs, 81% of participants report following the news most or some of the time, and 19% of participants report following only now and then or hardly at all. In this experiment, 44 participants failed the attention check and 8 participants withdrew. We do not find statistically significant differences in the failure rate on the attention check across political identities[105].

**Experiment 1b – March 19 2021 to June 30, 2022.** In Experiment 1b, 41,313 participants visited the experiment, passed the attention check, and provided 416,901 observations. According to data from Google Analytics, 76% of these participants participated from outside the United States. 5106 individuals participated in the experiment during the pre-registration window from March 4, 2021, to June 1, 2021. These participants found the website organically and completed 44,461 trials. Between June 1, 2021, and July 1, 2022, an additional 67,576 individuals (70% of whom visited from outside the United States) completed 566,343 trials. We include participants in Experiment 1b who participated outside the pre-registered window because we had an unexpectedly very large sample, which is due to around a thousand participants visiting the website each week and ten thousand participants visiting the website after it was posted to a website called Hacker News in March 2022. In total, 31,369 of these non-recruited participants failed the attention check.

**Experiment 2 – June 9, 2023.** In Experiment 2, we recruited 302 participants from the United States who successfully passed the attention check and provided 5964 observations. The demographic distribution of participants along sex, age, and ethnicity is: 54% male, 44% female, and 2% unknown; 46% 18 to 35, 46% 36−64, 5% over 64, and 2% unknown; and 70% White, 11% Black, 7% Asian, 5% mixed, 4% other, and 2% unknown. With respect to political beliefs, 63% of participants self-report their political preference as democratic, 16% as equally democratic and republican, and 22% as republican, similarly, 61% of participants report voting for Joseph Biden in 2020, 20% of participants report voting for Donald Trump in 2020, and the rest decline to answer or report voting for another candidate. In response to a pre-experiment question on participants' experience with deepfakes, fewer than 1% of participants responded that they have created their own deepfakes, 85% of participants have seen a few to several examples of deepfakes, and 13% of participants have yet to see their first deepfake or don't know their experience with deepfakes. In terms of trust and confidence in media, 49% of participants report a fair amount or a great deal of trust and confidence in media and 51% of participants report not very much or none at all. On the topic of following news on government and public affairs, 69% of participants report following the news most or some of the time, and 31% of participants report following only now and then or hardly at all. In this experiment, 30 participants fail the attention check.

**Experiment 3 – June 15, 2023 to June 20, 2023.** In Experiment 3, we recruited 1006 participants from the United States who successfully passed the attention check and provided 19,812 observations. The demographic distribution of participants along sex, age, and ethnicity is: 48% male, 51% female, and 1% unknown; 33% 18 to 35, 54% 36−64, 13% over 64, and 1% unknown; and 77% White, 13% Black, 6% Asian, 2% mixed, and 2% other. With respect to political beliefs, 62% of participants self-report their political preference as democratic, 12% as equally democratic and republican, and 26% as republican, similarly, 59% of participants report voting for Joseph Biden in 2020, 22% of participants report voting for Donald Trump in 2020, and the rest decline to answer or report voting for another candidate. In response to a pre-experiment question on participants' experience with deepfakes, fewer than 1% of participants responded that they have created their own deepfakes, 88% of participants have seen a few to several

examples of deepfakes, and 13% of participants have yet to see their first deepfake or don't know their experience with deepfakes. In terms of trust and confidence in media, 45% of participants report a fair amount or a great deal of trust and confidence in media and 55% of participants report not very much or none at all. On the topic of following news on government and public affairs, 75% of participants report following the news most or some of the time, and 25% of participants report following only now and then or hardly at all. In this experiment, 59 participants fail the attention check.

**Experiment 4 – June 14, 2023**. In Experiment 4, we recruited 206 participants from the United States who successfully passed the attention check and provided 3215 observations. The demographic distribution of participants along sex, age, and ethnicity is: 49% male, 48% female, and 3% unknown; 38% 18 to 35, 52% 36–64, 5% over 64, and 5% unknown; and 74% White, 7% Black, 8% Asian, 4% mixed, 2% other, and 3% unknown. With respect to political beliefs, 65% of participants self-report their political preference as democratic, 11% as equally democratic and republican, and 24% as republican, similarly, 58% of participants report voting for Joseph Biden in 2020, 21% of participants report voting for Donald Trump in 2020, and the rest decline to answer or report voting for another candidate. In response to a pre-experiment question on participants' experience with deepfakes, 1% of participants responded that they have created their own deepfakes, 87% of participants have seen a few to several examples of deepfakes, and 12% of participants have yet to see their first deepfake or don't know their experience with deepfakes. In terms of trust and confidence in media, 39% of participants report a fair amount or a great deal of trust and confidence in media and 61% of participants report not very much or none at all. On the topic of following news on government and public affairs, 76% of participants report following the news most or some of the time, and 24% of participants report following only now and then or hardly at all. In this experiment, 14 participants failed the attention check and 1 participant withdrew.

**Experiment 5 – December 22, 2023**. In Experiment 5, we recruited 200 participants from the United States who successfully passed the directed choice attention check and provided 2200 observations. The demographic distribution of participants along sex, age, and ethnicity is: 47.5% male, 50.5% female, and 2% unknown; 46% 18 to 35, 46% 36–64, 7% over 64, and 1% unknown; and 55% White, 18% Black, 11% Asian, 6% mixed, 6.5% other, and 3.5% unknown. In this experiment, 3 participants were excluded (2 participants failed the directed choice attention check, and 1 participant responded with the same unrelated response to all questions) and 21 participants withdrew before completing the experiment.

## Randomization
In all experiments, we randomized the order of the political speeches and each participant encounters each political speech only once. In Experiment 1, participants engaged with up to 32 unique political speeches. We randomized the display of the political speech as one of the seven modality conditions. In Experiment 2, participants engaged with a random sample of 20 unique videos from a pool of 60 videos, which consisted of 4 videos from each of the following 5 stimuli categories: 16 real videos from the PDD dataset, 16 deepfakes with audio from voice actors from the PDD dataset, 16 deepfakes with audio from a text-to-speech algorithm from the PDD dataset, 6 real videos used in Barari et al. 2021, and 6 deepfakes used in Barari et al. 2021[33]). In Experiment 3, participants engaged with a random sample of 20 unique political speeches from a pool of 32 unique political speeches, which consisted of 16 real videos and 16 deepfakes with audio from a text-to-speech algorithm from the PDD dataset. We randomized the display of the political speeches as one of the four modality conditions, and we also randomized the base rate of deepfakes seen. In Experiment 4, participants engaged with 32 unique political speeches, which

consisted of 16 real videos with the original audio and 16 real videos with voice actor audio. Finally, in Experiment 5, participants saw 10 stimuli randomly sampled from the same set as in Experiment 3.

## Reporting summary
Further information on research design is available in the Nature Portfolio Reporting Summary linked to this article.

## Data availability
The participant response data generated in this study have been deposited on Zenodo at https://doi.org/10.5281/zenodo.13340207[106]. The stimuli data are available under restricted access due to the sensitivity of the topic and potential misuse of stimulus materials, access can be obtained upon request to registered Zenodo users at https://doi.org/10.5281/zenodo.12709664[107].

## Code availability
All code produced to analyze the data generated in this study is available on Zenodo at https://doi.org/10.5281/zenodo.13340207[106].

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

## Acknowledgements
The authors would like to acknowledge support for signing videos from Truepic and $7000 in funding for participant recruitment from Truepic for Experiments 2 through 5, funding from MIT Media Lab member companies, and Kellogg School of Management, thanks Colin Cassidy, J-L Cauvin, Austin Nasso for providing voice impressions for stimuli, thank the following users who contributed sounds for stimuli from Freesound.org including aaronstar, aleclubin, cmilan, funwithsound, jgarc, johnsonbrandediting, klankbeeld, macohibs, mzui, noisecollector, peridactyloptrix, speedygonzo, zabuhailo, thank David Rand, Gordon Pennycook, Rahul Bhui, Yunhao (Jerry) Zhang, Ziv Epstein, and members of the Affective Computing lab at the MIT Media Lab and the Human Cooperation lab at MIT Sloan School of Management for helpful feedback on early versions of this manuscript, Anna Murphy, Shreya Kalyan, Theo Chen, and Alicia Guo for research assistance, and Craig Ferguson for feedback on hosting the experiment.

## Author contributions
M.G. conceived the experiments, A.S. and D.K. curated and created the deepfakes, N.S. performed audio engineering, A.S., M.G., and N.S. conducted the experiments, A.S. and M.G. analyzed the results, A.S., M.G., and N.S. wrote the manuscript, and A.S., A.L., M.G., N.S and R.P. reviewed and edited the manuscript.

## Competing interests
The authors declare no competing interests.
