## [Peer Review File · Nature Communications]

Human detection of political speech deepfakes across transcripts, audio, and videoReviewers' comments:

Reviewer #1 (Remarks to the Author):

Review of NCOMMS-22-37488

Reviewers: Stephan Lewandowsky and Simon Clark

Results and overall recommendation

This study tested 41,822 participants' ability to discern real and fake political speeches, when presented as video, audio, text, or combinations thereof. It further explored the impact of political alignment and cognitive reflection on accurate identification of fake and real speeches. Participants were presented with 32 excerpts of political speeches (half of which were fake) from the Presidential Deepfake Dataset, and were asked to indicate their confidence in whether the speech was fabricated. It was found that participants were more accurate when watching and hearing speeches, compared with just reading a transcript. The authors argue that this reflects a judgement based more on how the politician was speaking (i.e., audio-visual cues) than on what was said. The authors also found that accuracy was moderated by performance on the cognitive reflection task, arguing that a lack of cognitive reasoning leads to over-reliance on content.

This is a well-written manuscript. The dense introduction covers a lot of ground in introducing an important issue. The study is simple and elegantly designed, with a large sample size, and the analysis is competently conducted. It is, to our knowledge, the first study to focus primarily on the effect of modality on human detection of deepfake political speeches, and to test so many variations of modality. We are happy to endorse publication subject to a revision addressing the points below.

Major issues

>> possible training artifacts. Participants were given feedback after each stimulus and were exposed to 32 trials. This afforded considerable opportunity for learning. The data should be analyzed to estimate the effect of learning as this may crucially affect the aggregate results reported here. Figure 9 presents a precursor to this analysis but reports only guessing rates, not accuracy.

>> nomenclature. On occasion the authors refer to "fake" in the context of misinformation (e.g., headlines that are false). This is confusing in the present context where "fake" mainly refers to manipulated audio-visual material. It would be advisable to straighten out the terminology.

Detailed comments

[page#]:[para#]:[#line]

1:2:11 The use of "fake" in this context is problematic: the videos in India were not (deep)fakes but perfectly real videos that displayed an event other than the one claimed.

1:3:3 Whilst the authors' claim that "there exists little direct evidence for this [realism heuristic] applied to algorithmically manipulated videos" is broadly accurate, they may wish to mention the null effect of modality found in deepfake-specific studies by Hameleers et al.

(2022) (included at 2:2:12 in another context) and Murphy & Flynn (2021), in addition to Barari et al. (2021) cited here.

2:4 In general, the description of the method could be improved. For example, participants were asked to “discern truth from falsehood”; but what does that actually mean? Is this a fake/not fake discrimination task? Some more clarity would be advisable here.

2:4:3 It might be good to include (here or in the footnote) a brief explanation of why so many more participants found the questionnaire organically after the pre-registered cut-off date, than were part of the planned recruitment on Prolific (e.g., was it unexpectedly promoted on a popular website, or went viral?).

3:2:1 The authors state that political identity was balanced for 509 recruited participants, but there is no mention of this for the much larger number of 41,313 non-recruited participants. If political identity was not measured for non-recruited participants then it would be good to say so explicitly.

4:1:1 It might be preferable to refer to “the outcome measures” rather than listing them all here in the text, where it makes for cumbersome reading.

5:1 The results are difficult to interpret because the reader does not know on what basis a transcript could be identified as fake: was it the mismatch between expectation of what should be said by the politician and what the transcript showed? It might be helpful to provide a table before the results are presented that summarizes all available cues in each modality.

5:4 Singular of stimuli is “stimulus”.

7:3:5 In quoting Wittenberg et al. (2021), that people “are more likely to believe an event occurred when it is presented in video versus textual form”, the authors should perhaps acknowledge that this study nevertheless found only a minimal effect on persuasiveness, in line with the deepfake-specific study by Barari et al. (2021).

8:1:6 The authors argue that “low performance on silent videos offers evidence that visual artefacts and inconsistencies created by the lip syncing deepfake manipulations are not readily apparent to most people”. However, lip movement which is out of sync with the audio is one such potential inconsistency (arguably the most immediately apparent), and this would not be evident on a silent video.

8:4:8 It is not immediately clear why asymmetry in response times contributes to the “liar’s dividend”. Perhaps the authors could explain this point.

9:2:4 Again, the use of “fake” here is confusing: the papers cited did not consider deep fakes at all –they simply looked at information that was true or false, which is an entirely different issue because it is written content that is false and no perceptual cues are being altered.

9:6:5 When introducing stimuli, participants were asked to “...share how confident you are

that it is fabricated”, which seems to imply that the video is indeed fabricated. Given the clarification which followed (“Remember half the media snippets we show are real and half are fabricated”), this was unlikely to affect results. However, in future studies it might be better to use more neutral phrasing, such as “...share whether you think the video is fabricated, and how confident you are in this judgement”.

10:1 It is not entirely clear how the discordant videos can be authentic. Did Biden and Trump really spend 21 seconds saying things that run counter to their known opinion makeup? This needs to be explained.

References

- Barari, S., Lucas, C., & Munger, K. (2021). Political deepfake videos misinform the public, but no more than other fake media. OSF Preprints.
- Hameleers, M., van der Meer, T. G., & Dobber, T. (2022). You won't believe what they just said! the effects of political deepfakes embedded as vox populi on social media. *Social Media + Society*, 8(3), 20563051221116346.
- Murphy, G., & Flynn, E. (2022). Deepfake false memories. *Memory*, 30(4), 480–492.
- Wittenberg, C., Tappin, B. M., Berinsky, A. J., & Rand, D. G. (2021). The (minimal) persuasive advantage of political video over text. *Proceedings of the National Academy of Sciences*, 118(47), e2114388118.

Reviewer #2 (Remarks to the Author):

The authors report results of a study investigating human detection of political deepfakes across transcripts, audio, and video. The main finding reported is that access to audio and video improves participants' accuracy in distinguishing real from fabricated political speeches. A number of other results are discussed in the paper including the finding that low performance on the Cognitive Reflection Test is associated with greater reliance on what is said than on the visual information.

The research investigates a timely and important topic; the Introduction does a nice job of stating the past research and motivating the need for research. The efforts to recruit a large sample is excellent, however, below I do raise some concerns about whether the sample size is somewhat at the expense of methodological rigor. The main concern I have, though, is with the research stimuli. Having viewed the videos using the links provided in the Presidential Deepfakes Dataset, in my opinion, the quality of the deepfakes could easily account for the effect reported in the paper. The voice and lip-sync are reasonably easy to detect as fake (probably even more so when presented in a series of videos which include real videos of each politician allowing a comparison with real voice/lip movements) which limits the usefulness of the results and also limits real-world applicability. Without using highly sophisticated deepfakes in the research the extent to which the results are representative of the kinds of deepfake we worry about in the real world is limited. Readers might take from these results that people can readily detect deepfakes, yet, when faced with a highly sophisticated deepfake this may not be the case at all. I also wondered whether the difference in deepfake quality might account for the different finding to that reported by Barari et al. (2021) where no differences in truth discernment rate across video, audio, and text were found. I provide a range of comments below and also extend on my concerns above.

More detailed comments:

- The current findings seem to be in contrast with those of Barari et al. (2021), I'd like to see a more detailed description of how the current study is different to that work. The two proposed possibilities for Barari et al.'s results seem applicable also to the current research, so it would be useful to gain a greater appreciation of what is it that sets the two apart, therefore, why the results of the two studies are contradictory. Could it, for example, be difference in deepfake quality?
- Why in the recruited sample didn't all participants complete the same number of trials (482 rather than 509)?
- Can you say more about the minimum number of stimuli participants responded to – perhaps some indication of the distribution. And did each participant see a representative mix of modality, politician, and concordant vs. discordant conditions?
- Please can you explain the benefit of combining accuracy and confidence to generate a weighted accuracy score. Why not use the correct/incorrect responses?
- The results are fairly dense in places, to ensure full appreciation of the findings, could these sections be more concise, or could some of the results that are less central to the study be moved to an appendix?
- In the Discussion, the authors say that “In particular, we provide corroborating evidence to the conventional wisdom around the “seeing is believing” narrative (the realism heuristic that suggests people will tend to trust video over text²⁸ and recent results showing people “are more likely to believe an event occurred when it is presented in video versus textual form”²⁷)” Given that audio/video was associated with improved accuracy relative to text for both real and fabricated I don't think this is a convincing claim. Somewhat sceptically, having watched the videos, I would suggest that the mouth movement and audio is quite clearly fake on some of the stimuli which could explain why people are reasonably good at detecting these as fabricated when this information is provided (and these fabricated videos stand out from many of the real videos). With this in mind, I am not convinced that the finding that people's accuracy in discriminating real from fake content when these visual and audio cues are available particularly noteworthy.
- Reading on, the authors provide the following justification “participants' low performance on silent videos offers evidence that visual artefacts and inconsistencies created by the lip syncing deepfake manipulations are not readily apparent to most people, and as such, these videos represent a reasonable stimuli set for examining how well people can distinguish real from fake videos” I'm not sure this provides sufficient justification for the quality of the stimulus set? And isn't it the combination of the lip movements along with the audio that provides the most obvious clue in determining video authenticity?
- It would be helpful to have access to the datasets and code as part of the peer review process.

I have a few comments/questions about the method:

- In the experimental user interface, there is an option to indicate “I have seen/heard/read this before”. Were responses excluded if this option was selected?
- Was the only attention check the one at the beginning or were they presented throughout?
- How are the different length videos accounted for?
- You can submit before video finishes – was this controlled for?
- Feedback is given after each response which means that participants are aware if they got the answer right or not. How does this affect the results?
- What is the rationale for telling participants that 50% of the content is fake? This paired with feedback after each trial could have a substantial effect on responding.
- How did the authors ensure participants did not use a search engine to check authenticity of any of the videos? In particular the real ones which I assume can be found relatively easily on YouTube or a similar platform.

Minor comments:

- The authors state “We do not find demographic differences in recruited participants who passed the attention check.” Can they clarify where there are no demographic differences? Is it no difference between those who did and did not pass the attention check? Or something else?
- It seemed that those failing the attention check were excluded from any analyses but then reading on, it's become slightly unclear to me if unrecruited participants who failed the attention check were included or excluded from the analysis given the demonstration in the Supplementary Information that the results are robust with and without exclusions. Please confirm if the analysis in the main paper is based only on those who passed the attention check.
- Figure 2 – please make x-axis consistent in c and d to allow comparison. What are the error bars?

Reviewer #3 (Remarks to the Author):

The submitted manuscript examines people's ability to detect fabricated political content in text, audio, and video form. The comparison is useful but the article does not follow its preregistration and thus I have to recommend that it be declined. My comments follow.

1. The article refers to a preregistration and the deadline specified in the preregistration but otherwise seems to ignore the preregistration. This is not how preregistration works! The authors must either follow the preregistration or document the ways in which they deviate from it clearly so the reader knows what choices were deviations and the justifications for

those choices. Articles that cite a preregistration but do not follow it can mislead the reader into thinking the findings are confirmatory - 99.9% of readers will not look up the preregistration. Perhaps the authors have an elaborate accounting of the ways that what is reported in the paper corresponds to what is in the preregistration, but it is not obvious or transparent to me - for instance, the hypotheses from the preregistration aren't mentioned, the discordancy concept isn't in the preregistration, the reported exclusion rules differ, etc. etc.

2. Bonferonni is arguably too conservative and controls the FWER, which is arguably less relevant to most research questions and contexts than the FDR <https://egap.org/resource/10-things-to-know-about-multiple-comparisons/>. I would suggest the sharpened Benjamini-Hochberg (Benjamini, Krieger, and Yekutieli (2006)).

3. Telling the participants the base rates of false content (Figure 5) seems highly problematic for measuring real-world behavior because (a) people don't know these and often have wildly inaccurate guesses and (b) differences in perceived base rates by content type may drive differences in real-world performance. The authors' design allows us to isolate the modality effect by holding base rates fixed, but they should discuss how we should extrapolate their findings to the real world given (a) and (b) above. This is not mentioned until the conclusion.

4. The authors should test for non-linearity in moderation by CRT per Heinmueller et al. (2019).

5. The authors are incorrectly plotting interaction terms as treatment effects in Figure 4. That is a coefficient plot and very difficult to interpret. I recommend a proper marginal effects plot over the range of CRT for different types of content.

Reviewer #4 (Remarks to the Author):

I enjoyed reading this manuscript, in which the authors report the findings of an innovative experiment meant to distill differences in the detection of fabrication content across different media modes (text, audio, video). The authors created ~ 32*7 stimuli (the exact number was not entirely clear to me), which is impressive and laudable. I think the authors have certainly executed a good study, but I think the scope is much too focused for this journal.

This immediately brings me to my biggest point of critique. This experiment creates a rather artificial setting by explicitly asking people to rate the accuracy of the stimuli, and by telling the participants repeatedly that half of the stimuli are real, and half are fabricated. The authors briefly mention this in the discussion section, but the way this experiment is designed causes participants to take on a very unnatural position. A very skeptical position, in which they know that half of the content they see is fake, and half is real. In reality this never happens, and in reality most people rarely encounter fake content. By no means does this mean that the experiment is not useful or informative. However, it does mean that the findings do not reveal much about how people in reality will respond to fabricated (or real)

content.

Another problem that adds to the artificiality of this experiment is that users were confronted in the experiment with the real voices of Trump and Biden as well as the imitated voices of both. If you first hear the real voice of Biden, and a moment later you listen to a voice actor it is much easier to spot the fabricated content than when you did not just hear Biden speak. I wonder how good the audio and video + audio would score in a separate experiment. Were the stimuli pre-tested?

At the very least there should be an additional experiment, in which people are much less primed to detect fabricated content (and in which they do not know the baseline of fabricated content). An additional experiment is recommended also, because the findings are based upon only one experiment. Of course there were many additional participants from across the globe who stumbled upon the website, but these participants can (and were) only used for robustness purposes in a limited way (for one, because they are intrinsically motivated to participate). Thus, as we know, we have only one experiment that is quite limited in scope and we cannot yet know how robust the findings really are.

Finally, I enjoyed reading in the discussion section authors' reflections on how each layer of information provides the participant with more information to judge the veracity of a stimulus. In fact, with each layer (except subtitles, which is also written text!), participants' accuracy increases. For text, participants did not do well in spotting the fabricated (and non-fabricated) content. Add audio and they score much better. Add video and audio and they score even better. Authors conclude that "Overall, the experiment's results show that as participants have access to more information via audio and video, they are better able to distinguish whether a political speech has been fabricated". But this conclusion is based on an experiment with a very limited scope. The conclusion should be: If you tell participants that half of the stimuli they get displayed is fabricated, then more information via audio and video means better accuracy ratings.

Taken together, I think the authors did a good job but the study is more fitting for a journal with a more focused scope.

Responses to R1

Review of NCOMMS-22-37488

Reviewers: Stephan Lewandowsky and Simon Clark

Results and overall recommendation

This study tested 41,822 participants' ability to discern real and fake political speeches, when presented as video, audio, text, or combinations thereof. It further explored the impact of political alignment and cognitive reflection on accurate identification of fake and real speeches. Participants were presented with 32 excerpts of political speeches (half of which were fake) from the Presidential Deepfake Dataset, and were asked to indicate their confidence in whether the speech was fabricated. It was found that participants were more accurate when watching and hearing speeches, compared with just reading a transcript. The authors argue that this reflects a judgement based more on how the politician was speaking (i.e., audio-visual cues) than on what was said. The authors also found that accuracy was moderated by performance on the cognitive reflection task, arguing that a lack of cognitive reasoning leads to over-reliance on content.

This is a well-written manuscript. The dense introduction covers a lot of ground in introducing an important issue. The study is simple and elegantly designed, with a large sample size, and the analysis is competently conducted. It is, to our knowledge, the first study to focus primarily on the effect of modality on human detection of deepfake political speeches, and to test so many variations of modality. We are happy to endorse publication subject to a revision addressing the points below.

We are deeply appreciative of R1's recognition of the subject matter's importance, the manuscript's clarity, the experimental design's simplicity, elegance, and size, the analysis' competence, and the novelty of the study!

Major issues

>> possible training artifacts. Participants were given feedback after each stimulus and were exposed to 32 trials. This afforded considerable opportunity for learning. The data should be analyzed to estimate the effect of learning as this may crucially affect the aggregate results reported here. Figure 9 presents a precursor to this analysis but reports only guessing rates, not accuracy.

Thank you for raising concerns about potential effects of learning over time. In the results section, we now report the minimal learning effects in experiment 1a (where we provide feedback), which do not affect the overall results. Also, we report that we do not find learning effects in experiments 2 or 3 (where we do not provide feedback until the debrief at the end of the experiment). We also find minor learning effects in experiment 4 on the voice actor audio

versus actual speaker audio task (where we do not provide feedback until the end of the experiment).

>> nomenclature. On occasion the authors refer to “fake” in the context of misinformation (e.g., headlines that are false). This is confusing in the present context where “fake” mainly refers to manipulated audio-visual material. It would be advisable to straighten out the terminology.

We agree, and we revised the manuscript such that we use more nuanced nomenclature (e.g. false news, false news headlines, misleading) and “fake” refers precisely to audio-visual manipulations only.

Detailed comments

[page#]:[para#]:[#line]

1:2:11 The use of “fake” in this context is problematic: the videos in India were not (deep)fakes but perfectly real videos that displayed an event other than the one claimed.

We agree, and we changed this to “misleading.”

1:3:3 Whilst the authors’ claim that “there exists little direct evidence for this [realism heuristic] applied to algorithmically manipulated videos” is broadly accurate, they may wish to mention the null effect of modality found in deepfake-specific studies by Hameleers et al. (2022) (included at 2:2:12 in another context) and Murphy & Flynn (2021), in addition to Barari et al. (2021) cited here.

Thank you for this suggestion! We have added a citation to Murphy and Flynn, and we edited the manuscript to include all three of these citations in this same paragraph.

2:4 In general, the description of the method could be improved. For example, participants were asked to “discern truth from falsehood”; but what does that actually mean? Is this a fake/not fake discrimination task? Some more clarity would be advisable here.

Excellent question! We clarify the exact wording of the task in experiment 1a and 1b in the Methods section: “In the experiment, we asked participants to identify fabricated and non-fabricated stimuli. After collecting informed consent and presenting participants with instructions, we show participants a short political speech and ask “Did [Joseph Biden/Donald Trump] say that?” followed by “Please [read/listen/watch] this [transcript/audio clip/video] from [Joseph Biden/Donald Trump] and share how confident you are that it is fabricated. Remember half the media snippets we show are real and half are fabricated.” Figure 8 in the Supplementary Information section presents a screenshot of the user interface, which shows participants were instructed to move a slider to report their confidence from 50% to 100% that a stimulus is fabricated (or 50% to 100% that a stimulus is not fabricated).”

2:4:3 It might be good to include (here or in the footnote) a brief explanation of why so many more participants found the questionnaire organically after the pre-registered cut-off date, than were part of the planned recruitment on Prolific (e.g., was it unexpectedly promoted on a popular website, or went viral?).

Thank you for this suggestion. In the “Participants” section in the Methods, we include the following: “We include participants in experiment 1b who participated outside the pre-registered

window because we had an unexpectedly very large sample, which is due to around a thousand of participants visiting the website each week and over ten thousand participants visiting the website after it was posted to a website called Hacker News in March 2022.”

3:2:1 The authors state that political identity was balanced for 509 recruited participants, but there is no mention of this for the much larger number of 41,313 non-recruited participants. If political identity was not measured for non-recruited participants then it would be good to say so explicitly.

We agree, and we edited the text to make this explicit: “In experiment 1b, we collected data from participants who visited the experiment organically but did not collect demographic, political identity, or pre-experiment questions for these non-recruited participants.”

4:1:1 It might be preferable to refer to “the outcome measures” rather than listing them all here in the text, where it makes for cumbersome reading.

We agree, and we changed the text accordingly.

5:1 The results are difficult to interpret because the reader does not know on what basis a transcript could be identified as fake: was it the mismatch between expectation of what should be said by the politician and what the transcript showed? It might be helpful to provide a table before the results are presented that summarizes all available cues in each modality.

This is an excellent question, and we describe the available sources of information as “the content of what is said and the audio-visual cues as to how it is said.” The text contains the content of what is said but not how it is said, the silent video without subtitles in experiment 1 contains the visual component to how it is said, and the audio and video with audio contains both what is said and how it is said. There are many reasons why someone might identify a speech transcript as fake. For example the mismatch between the expectation of what they might expect the politician to say based on his or her views and what the transcript shows could be one reason. But there are many other reasons; maybe a participant thinks no one would say such a thing, maybe a participant thinks only democrats or republicans would say such a thing, maybe a participant thinks this doesn’t sound like the politician’s style, maybe a participant thinks the speech is inappropriate given the politician’s status, or maybe it’s something else. We leave out the particular reason a participant identifies text transcripts as fabricated or not fabricated because we do not ask participants for this. We note that participants are near random guessing at text transcripts (57%) which means the content is an indicator but not an obvious give away that a speech is real or fake.

5:4 Singular of stimuli is “stimulus”.

Great catch! We fixed this typo.

7:3:5 In quoting Wittenberg et al. (2021), that people “are more likely to believe an event occurred when it is presented in video versus textual form”, the authors should perhaps acknowledge that this study nevertheless found only a minimal effect on persuasiveness, in line with the deepfake-specific study by Barari et al. (2021).

We now added "(though note Wittenberg et al 2021 finds minimal effects of video on persuasiveness)"

8:1:6 The authors argue that "low performance on silent videos offers evidence that visual artefacts and inconsistencies created by the lip syncing deepfake manipulations are not readily apparent to most people". However, lip movement which is out of sync with the audio is one such potential inconsistency (arguably the most immediately apparent), and this would not be evident on a silent video.

Thank you for raising this point. We have removed this phrase, and we showcase the high quality of the deepfakes by showing participants are 72% accurate on deepfakes with text to speech audio. We write: "Participants are closer to random guessing than a perfect score on the enhanced PDD text-to-speech deepfakes but closer to a perfect score on the rest of the stimuli"

8:4:8 It is not immediately clear why asymmetry in response times contributes to the "liar's dividend". Perhaps the authors could explain this point.

For clarity and conciseness, we removed the point about asymmetry in response times, which we found is no longer the case once we control for video/audio duration.

9:2:4 Again, the use of "fake" here is confusing: the papers cited did not consider deep fakes at all –they simply looked at information that was true or false, which is an entirely different issue because it is written content that is false and no perceptual cues are being altered.

We agree, and we revised the manuscript such that we use more nuanced nomenclature (e.g. false news, false news headlines, misleading) and leave "fake" to refer precisely to audio-visual manipulations only.

9:6:5 When introducing stimuli, participants were asked to "...share how confident you are that it is fabricated", which seems to imply that the video is indeed fabricated. Given the clarification which followed ("Remember half the media snippets we show are real and half are fabricated"), this was unlikely to affect results. However, in future studies it might be better to use more neutral phrasing, such as "...share whether you think the video is fabricated, and how confident you are in this judgement".

Thank you for this suggestion. We adapted the new experiments to follow this wording and remove the clarification.

10:1 It is not entirely clear how the discordant videos can be authentic. Did Biden and Trump really spend 21 seconds saying things that run counter to their known opinion makeup? This needs to be explained.

We define "discordant" political speeches as political speeches that do not match what people believe are a politician's current views based on a pilot described in the Methods section where we instructed 84 participants "For each statement, we want you to rank how closely the statement matches your understanding of President Joseph Biden or President Donald Trump's political views" and asked them to provide a judgment on a 5-point Likert scale from "Strongly Disagree" (-2) to "Strongly Agree" (2) that "This statement matches President [Joseph Biden's/Donald Trump's] political viewpoint: [statement]". There are a number of reasons how

we could find these statements: (1) politicians change views over time (2) people have misconceptions of politicians' views (3) quotes taken out of context can appear to be counter to a politician's known views. Here's an example of one stimuli of Biden discussing abortion: "With regard to, um, with regard to abortion, I accept my church's position on abortion as a what we call, "De fide doctrine"". Life begins at conception and that's the church's judgment and I accept it in my personal life."

References

- Barari, S., Lucas, C., & Munger, K. (2021). Political deepfake videos misinform the public, but no more than other fake media. OSF Preprints.
- Hameleers, M., van der Meer, T. G., & Dobber, T. (2022). You won't believe what they just said! the effects of political deepfakes embedded as vox populi on social media. *Social Media + Society*, 8(3), 20563051221116346.
- Murphy, G., & Flynn, E. (2022). Deepfake false memories. *Memory*, 30(4), 480–492.
- Wittenberg, C., Tappin, B. M., Berinsky, A. J., & Rand, D. G. (2021). The (minimal) persuasive advantage of political video over text. *Proceedings of the National Academy of Sciences*, 118(47), e2114388118.

Responses to R2

The authors report results of a study investigating human detection of political deepfakes across transcripts, audio, and video. The main finding reported is that access to audio and video improves participants' accuracy in distinguishing real from fabricated political speeches. A number of other results are discussed in the paper including the finding that low performance on the Cognitive Reflection Test is associated with greater reliance on what is said than on the visual information.

The research investigates a timely and important topic; the Introduction does a nice job of stating the past research and motivating the need for research. The efforts to recruit a large sample is excellent, however, below I do raise some concerns about whether the sample size is somewhat at the expense of methodological rigor.

We appreciate R2's recognition of the topic's timeliness and importance and how the introduction nicely motivates the need for this research.

The main concern I have, though, is with the research stimuli. Having viewed the videos using the links provided in the Presidential Deepfakes Dataset, in my opinion, the quality of the deepfakes could easily account for the effect reported in the paper. The voice and lip-sync are reasonably easy to detect as fake (probably even more so when presented in a series of videos which include real videos of each politician allowing a comparison with real voice/lip movements) which limits the usefulness of the results and also limits real-world applicability. Without using highly sophisticated deepfakes in the research the extent to which the results are representative of the kinds of deepfake we worry about in the real world is limited. Readers might take from these results that people can readily detect deepfakes, yet, when faced with a highly sophisticated deepfake this may not be the case at all. I also wondered whether the difference in deepfake quality might account for the different finding to that reported by Barari et al. (2021) where no differences in truth discernment rate across video, audio, and text were found. I provide a range of comments below and also extend on my concerns above.

Thank you for raising concerns with the research stimuli. In order to address this main concern, we re-created the stimuli set, which we describe in the Experiment Stimuli section of the Methods: “The 16 enhanced deepfakes include source videos more amenable to deepfake lip-syncing manipulations than the initial source videos for the PDD deepfakes, visual touch ups with the deepface lab algorithm, audio from both a voice actor and a text-to-speech algorithm trained on the speakers’ voices, and additional audio engineering to add in background noise and subtle acoustic elements to make the audio appear real. We provide additional details on the deepfake generation process below in the section titled “Enhanced Deepfake Generation Process.” As a result, we have a stimuli set of the most highly sophisticated deepfakes that can be produced via today’s algorithms combined with the artistry of a voice actor and audio engineer. We note that we did not hire a visual effects studio to further enhance the deepfakes by a visual effects (VFX) artist. The cost of VFX artists from studios such as Revel.ai or Metaphysic estimate their prices at \$1,000 per second for creating deepfakes, and we have over 300 seconds of deepfake footage, which would cost \$300,000. As such, this cost would be prohibitive.

The enhanced text-to-speech PDD deepfake videos are accurately identified as fake in 72% of observations whereas the original PDD deepfake videos, the Barari et al 2021 videos, and the enhanced voice actor PDD videos are identified in the range of 83% to 87% of observations. We note: “Participants are closer to random guessing than a perfect score on the enhanced PDD text-to-speech deepfakes but closer to a perfect score on the rest of the stimuli.”

More detailed comments:

- The current findings seem to be in contrast with those of Barari et al. (2021), I’d like to see a more detailed description of how the current study is different to that work. The two proposed possibilities for Barari et al.’s results seem applicable also to the current research, so it would be useful to gain a greater appreciation of what is it that sets the two apart, therefore, why the results of the two studies are contradictory. Could it, for example, be difference in deepfake quality?

Figure 2 and Figure 4 reveal comparisons between discernment on the PDD videos (all speeches by Joseph Biden or Donald Trump) and the videos used in Barari et al 2021 (two deepfakes from SNL skits, two deepfakes explicitly mentioning misinformation, and two deepfakes that are clearly satire where Trump claims to have ended AIDS and Boris Johnson endorses his political opponent). Our enhanced text-to-speech deepfake videos (in experiments 2 and 3) are significantly harder for people to detect than the Barari et al 2021 videos and all the rest of our videos are, on average, at a similar level of difficulty for people to detect.

This study differs from Barari et al 2021 in a number of ways: (1) We focus the detection task on non-satire political speeches whereas Barari et al 2021 focus the task on mostly satirical deepfakes and gaffes and unintentional hot mics of politicians. (2) In experiment 1, we consider all 7 permutations of text, audio, and video while Barari et al considers only text vs. audio vs. video. By considering all permutations, we can look in granular detail at the content of what is said (text) vs. the perceptual cues to how it is said (silent video without text). In experiment 2, we compare our stimuli to Barari et al 2021 stimuli and show our stimuli are higher quality with respect to difficulty in detecting the stimuli as real/fake. In experiment 3, we provide a conceptual replication of experiment 1 with enhanced stimuli and additional randomized conditions to control for framing. In experiment 4, we examine people’s ability to distinguish voice actor audio from the real speaker’s audio. (3) Our stimuli set contains 32 political

speeches (16 real and 16 fake), which we created ourselves and Barari et al contains 9 real and 6 fake, which are drawn from the Internet and Agarwal et al 2019's deepfakes of SNL skits (4) We also note that Barari et al is a pre-print that has not yet been published in a peer-reviewed journal.

We also note that the current findings contrast in some ways and agree in others with Barari et al 2021. For example, our findings and Barari et al's findings both show that text is not more believable than video. In contrast, we find people are much better at distinguishing between true and false in video than text, which likely is due to the visual and audio elements of the speeches.

- Why in the recruited sample didn't all participants complete the same number of trials (482 rather than 509)?

We explain this in the Participants section in the Methods: "In experiments 1a, 2, 3, and 4, 94%, 83%, 89%, and 84% of participants provided responses to the complete set of stimuli, which results in 98.8%, 98.7%, 98.5%, and 97.6% of the expected data in each experiment. The missing data appears to be missing due to intermittent or slow network connection issues where participants could proceed without their data getting submitted to the server." In addition, 8 of the 509 participants requested to withdraw from the experiment (technically on Prolific this is called a "Returned Submission"), so we have excluded them in the analysis.

- Can you say more about the minimum number of stimuli participants responded to – perhaps some indication of the distribution. And did each participant see a representative mix of modality, politician, and concordant vs. discordant conditions?

Yes, each participant in experiment 1a is randomized to the modality in which they see the speech and 94% of participants see all 32 speeches (and most participants see most speeches). So, in experiment 1a, participants see a representative mix of modality and the full population of the politician and concordant/discordant conditions. This representativeness is true for all experiments.

- Please can you explain the benefit of combining accuracy and confidence to generate a weighted accuracy score. Why not use the correct/incorrect responses?

Thank you for raising this question. We use "weighted accuracy" or "confidence score" as we now call it because we pre-registered it following Groh et al 2022 PNAS because weighted accuracy offers more information than a binary correct/incorrect response. Ultimately, the results are qualitatively the same for weight accuracy and correct/incorrect responses, but weighted accuracy allows participants to share whether they are extremely confident that a stimulus is/is not fabricated or whether they are only slightly confident or whether they are completely unsure.

In experiments 2 to 4, we separate accuracy and confidence.

Also, we report both accuracy and confidence score for experiment 1a in the results now for clarity.

- The results are fairly dense in places, to ensure full appreciation of the findings, could these sections be more concise, or could some of the results that are less central to the study be moved to an appendix?

Thank you for this suggestion! We reflected on where the results are dense and adapted the manuscript accordingly to increase clarity and move portions to the appendix.

- In the Discussion, the authors say that “In particular, we provide corroborating evidence to the conventional wisdom around the “seeing is believing” narrative (the realism heuristic that suggests people will tend to trust video over text²⁸ and recent results showing people “are more likely to believe an event occurred when it is presented in video versus textual form”²⁷)” Given that audio/video was associated with improved accuracy relative to text for both real and fabricated I don’t think this is a convincing claim.

We have edited this text to increase clarity as follows: “In the context of authentic speeches, we provide corroborating evidence for the conventional wisdom around the “seeing is believing” narrative (the realism heuristic that suggests people will tend to trust video over text and recent results showing people ‘are more likely to believe an event occurred when it is presented in video versus textual form’); people are significantly more accurate at identifying authentic speeches as authentic when the speeches include audio and visual modalities as opposed to only text (although note Wittenberg et al 2021 finds minimal effects of video on persuasiveness). In contrast when considering fabricated speeches, these results add considerable nuance to the seeing is believing narrative: people are significantly more accurate at identifying fabricated speeches as fabricated when the speeches include audio and visual modalities as opposed to only text. In other words, we find participants are significantly more accurate at distinguishing between authentic and fabricated political videos than transcripts.”

Somewhat sceptically, having watched the videos, I would suggest that the mouth movement and audio is quite clearly fake on some of the stimuli which could explain why people are reasonably good at detecting these as fabricated when this information is provided (and these fabricated videos stand out from many of the real videos). With this in mind, I am not convinced that the finding that people’s accuracy in discriminating real from fake content when these visual and audio cues are available particularly noteworthy.

We appreciate R2’s skepticism that results are based on clearly fake mouth movements and audio, and as such, we re-created and enhanced all stimuli accordingly. Also, we note that participants may not know to focus solely on the mouth when trying to distinguish real from fabricated. Participants’ accuracy is 72% in experiment 2, which suggests that people generally had trouble with discernment and are closer to random guessing than a perfect score.

- Reading on, the authors provide the following justification “participants’ low performance on silent videos offers evidence that visual artefacts and inconsistencies created by the lip syncing deepfake manipulations are not readily apparent to most people, and as such, these videos represent a reasonable stimuli set for examining how well people can distinguish real from fake videos” I’m not sure this provides sufficient justification for the quality of the stimulus set? And isn’t it the combination of the lip movements along with the audio that provides the most obvious clue in determining video authenticity?

Thank you for raising this point, and yes the combination of the lip movements along with the audio is one clue in determining video authenticity. We have removed this phrase, and we showcase the high quality of the deepfakes by showing participants are 72% accurate on

deepfakes with text to speech audio. We write: “Participants are closer to random guessing than a perfect score on the enhanced PDD text-to-speech deepfakes but closer to a perfect score on the rest of the stimuli”

- It would be helpful to have access to the datasets and code as part of the peer review process.

We provide the data and code on Research Box at https://researchbox.org/1723&PEER_REVIEW_passcode=EGVULE, which is now linked to in the manuscript.

I have a few comments/questions about the method:

- In the experimental user interface, there is an option to indicate “I have seen/heard/read this before”. Were responses excluded if this option was selected?

In experiments 1a, 2, 3, and 4, the percent of observations where participants have seen a stimulus before is 3, 4, 6, 3, and 7, respectively. We conduct robustness checks: the results of main findings hold and there is no significant difference between the results with and without people who check this box. We do not drop responses because we did not pre-register dropping these responses and the results do not change. Furthermore, we find participants are similarly accurate on real videos they claim to have seen before as real videos they haven’t seen before, and participants are significantly less accurate on fake videos they report to have seen before, which suggests some people report they have seen a video before but they had not.

- Was the only attention check the one at the beginning or were they presented throughout?

Yes, there is only one attention check at the beginning of the experiment.

- How are the different length videos accounted for?

The videos are very similar lengths, and we measure response time controlling for video duration. In particular, the interquartile range of video durations is 18.99 seconds to 21.54 seconds, the minimum duration is 12.90 seconds and the maximum is 30.72 seconds.

- You can submit before video finishes – was this controlled for?

In experiment 1, participants can submit responses after they watch at least 7 seconds of a video. In experiments 2, 3, and 4, participants can submit responses after they watch at least 15 seconds of a video. We find that participants, on average, take an additional 8 seconds beyond the video duration to respond. We also note it’s certainly possible to distinguish real from fake in the first few seconds of watching videos. In experiment 2, we find 9 percent of observations involve participants responding before a video has finished and the minimum time taken by a participant is 1.7 seconds before a video has finished.

- Feedback is given after each response which means that participants are aware if they got the answer right or not. How does this affect the results?

In experiment 1, we give feedback after each response to inform and engage participants. In experiments 2, 3, and 4, we do not give feedback to participants until the debrief at the end of

the experiment to address this important question of how feedback affects the results. In experiment 1, we find minimal effects of learning over time (0.12 percentage point accuracy increase for each stimuli seen) and we do not find learning effects in experiment 2 or 3, and we find minimal learning effects in experiment 4.

- What is the rationale for telling participants that 50% of the content is fake? This paired with feedback after each trial could have a substantial effect on responding.

In experiment 1, we informed participants of the distribution of real and fake content such that participants' priors on baserates are all the same. In order to address concerns around base rates, we did not inform participants of the base rates in experiments 2, 3, and 4.

- How did the authors ensure participants did not use a search engine to check authenticity of any of the videos? In particular the real ones which I assume can be found relatively easily on YouTube or a similar platform.

Thank you for raising this question. We cannot ensure participants did not use a search engine to check the authenticity of any of the videos, but it turns out it's not trivial to find many of these political speeches. The response times suggest people did not spend the extra time and effort to try to search for these.

Minor comments:

- The authors state "We do not find demographic differences in recruited participants who passed the attention check." Can they clarify where there are no demographic differences? Is it no difference between those who did and did not pass the attention check? Or something else?

Thank you for highlighting this lack of clarity! We edited the sentence as follows: "We do not find statistically significant differences in the failure rate on the attention check across political identities."

- It seemed that those failing the attention check were excluded from any analyses but then reading on, it's become slightly unclear to me if unrecruited participants who failed the attention check were included or excluded from the analysis given the demonstration in the Supplementary Information that the results are robust with and without exclusions. Please confirm if the analysis in the main paper is based only on those who passed the attention check.

We have edited the paper and clarified that all results in the paper are from participants who passed the attention check.

- Figure 2 – please make x-axis consistent in c and d to allow comparison. What are the error bars?

We clarify that error bars represent 95% confidence intervals, and we removed Figures 2c and 2d and added new Figures to demonstrate results from the additional 3 experiments we ran.

The submitted manuscript examines people's ability to detect fabricated political content in text, audio, and video form. The comparison is useful but the article does not follow its preregistration and thus I have to recommend that it be declined. My comments follow.

1. The article refers to a preregistration and the deadline specified in the preregistration but otherwise seems to ignore the preregistration. This is not how preregistration works! The authors must either follow the preregistration or document the ways in which they deviate from it clearly so the reader knows what choices were deviations and the justifications for those choices. Articles that cite a preregistration but do not follow it can mislead the reader into thinking the findings are confirmatory - 99.9% of readers will not look up the preregistration. Perhaps the authors have an elaborate accounting of the ways that what is reported in the paper corresponds to what is in the preregistration, but it is not obvious or transparent to me - for instance, the hypotheses from the preregistration aren't mentioned, the discordancy concept isn't in the preregistration, the reported exclusion rules differ, etc. etc.

We appreciate R3's concern that we clarify what is pre-registered and what is not. We certainly do not intend or want to mislead readers. We address R3's concern by (1) clarifying what is pre-registered and what is not in the original experiment (2) pre-registering a second, third, and fourth experiment. At the beginning of the results section, we write "Experiments 1a, 2, 3, and 4 involve participants recruited from Prolific and are pre-registered on aspredicted.org at the following URLs: 1a, 2, 3, and 4 [links provided in the paper]. Experiment 1b is not pre-registered and includes 41,313 participants who discovered the experiment through search engines or the news."

2. Bonferroni is arguably too conservative and controls the FWER, which is arguably less relevant to most research questions and contexts than the FDR <https://egap.org/resource/10-things-to-know-about-multiple-comparisons/>. I would suggest the sharpened Benjamini-Hochberg (Benjamini, Krieger, and Yekutieli (2006)).

Thank you for the advice on how to more effectively address multiple hypothesis testing by focusing on the False Discovery Rate with the Benjamini-Hochberg method. We follow this advice, pre-register the B-H adjustments for the families of multiple comparisons, and present these in the tables in the appendix.

3. Telling the participants the base rates of false content (Figure 5) seems highly problematic for measuring real-world behavior because (a) people don't know these and often have wildly inaccurate guesses and (b) differences in perceived base rates by content type may drive differences in real-world performance. The authors' design allows us to isolate the modality effect by holding base rates fixed, but they should discuss how we should extrapolate their findings to the real world given (a) and (b) above. This is not mentioned until the conclusion.

Thank you for raising these two important points. In order to address concerns about sharing the base rate of false content, we conducted multiple follow up experiments where we do not tell participants the base rate of false content. In experiment 2, 60% of the content is fabricated. And in experiment 3, we have two conditions that we randomly assign participants to where 20% of the content is fabricated and where 80% of the content is fabricated. In experiment 3 (where we do not mention the base rates) we find participants guess fake 50% of the time for video with audio, 49% of the time for audio only or silent video with text, and 45% of the time for text transcripts.

4. The authors should test for non-linearity in moderation by CRT per Heinmueller et al. (2019).

Thank you pointing us to Hainmueller et al's work on multiplicative interaction models. In an effort to focus on the main analysis and the additional experiments where we do not include a measure of the CRT, we have removed the analysis of heterogeneous treatment effects from the paper.

5. The authors are incorrectly plotting interaction terms as treatment effects in Figure 4. That is a coefficient plot and very difficult to interpret. I recommend a proper marginal effects plot over the range of CRT for different types of content.

Thank you for the suggestion. We have removed the interaction with CRT from the paper to increase clarity and focus on the main results from the additional experiments 2-4.

Responses to R4

I enjoyed reading this manuscript, in which the authors report the findings of an innovative experiment meant to distill differences in the detection of fabrication content across different media modes (text, audio, video). The authors created ~ 32*7 stimuli (the exact number was not entirely clear to me), which is impressive and laudable. I think the authors have certainly executed a good study, but I think the scope is much too focused for this journal.

We appreciate R4's characterization of this study as innovative and well-executed. And, we confirm that R4 is correct that the original stimuli set is composed of 32*7 stimuli made up of 32 political speeches across 7 combinations of text, audio, and video.

This immediately brings me to my biggest point of critique. This experiment creates a rather artificial setting by explicitly asking people to rate the accuracy of the stimuli, and by telling the participants repeatedly that half of the stimuli are real, and half are fabricated. The authors briefly mention this in the discussion section, but the way this experiment is designed causes participants to take on a very unnatural position. A very skeptical position, in which they know that half of the content they see is fake, and half is real. In reality this never happens, and in reality most people rarely encounter fake content. By no means does this mean that the experiment is not useful or informative. However, it does mean that the findings do not reveal much about how people in reality will respond to fabricated (or real) content.

We appreciate your critique that the baseline rate of false news is relatively low (as we discuss in the Discussion section) and people are generally not primed to be skeptical (although see Pennycook et al 2021 that shows prompting people to think about accuracy reduces the sharing of false news and Epstein et al 2023 that shows prompting people to think about sharing reduces accuracy on discerning false news). In the future, people may potentially encounter false news very frequently and the base-rates may change, so we believe addressing the fundamental question of this experiment of how well people distinguish between authentic and fabricated content across media modalities may also directly address future real-world concerns. In order to address concerns around the base-rate, we run additional experiments including experiment 3 where we do not share the base rate and randomize participants to a low or high base rate of fake news.

Another problem that adds to the artificiality of this experiment is that users were confronted in the experiment with the real voices of Trump and Biden as well as the imitated voices of both. If you first hear the real voice of Biden, and a moment later you listen to a voice actor it is much easier to spot the fabricated content than when you did not just hear Biden speak. I wonder how good the audio and video + audio would score in a separate experiment. Were the stimuli pre-tested?

Thank you for raising this question about pre-testing the stimuli and how accuracy may depend on the context of hearing Trump's real voice in one video and the voice actor's voice in another. We conducted and ran experiment 2 to provide insight into how accurately people can distinguish real and deepfake videos of Trump and Biden (and other videos used in Barari et al 2021) and we find the enhanced text-to-speech PDD deepfake videos are accurately identified as fake in 72% of observations whereas the original PDD deepfake videos, the Barari et al 2021 videos, and the enhanced voice actor PDD videos are identified in the range of 83% to 87% of observations. We note: "Participants are closer to random guessing than a perfect score on the enhanced PDD text-to-speech deepfakes but closer to a perfect score on the rest of the stimuli." We do not find participants increase their accuracy as they see more videos, which suggests that first hearing the real voice of Biden and then the voice actor's does not appear to influence the results. Also, we conceptually replicate experiment 1 with experiment 3 using the text-to-speech deepfakes.

At the very least there should be an additional experiment, in which people are much less primed to detect fabricated content (and in which they do not know the baseline of fabricated content). An additional experiment is recommended also, because the findings are based upon only one experiment. Of course there were many additional participants from across the globe who stumbled upon the website, but these participants can (and were) only used for robustness purposes in a limited way (for one, because they are intrinsically motivated to participate). Thus, as we know, we have only one experiment that is quite limited in scope and we cannot yet know how robust the findings really are.

Thank you for the suggestion of running an additional experiment. In order to address concerns about sharing the base rate of false content, we conducted three follow up experiments where we did not tell participants the base rate of false content. In experiment 2, 60% of the content is fabricated. And in experiment 3, we have two conditions that we randomly assign participants to where 20% of the content is fabricated and where 80% of the content is fabricated.

Finally, I enjoyed reading in the discussion section authors' reflections on how each layer of information provides the participant with more information to judge the veracity of a stimulus. In fact, with each layer (except subtitles, which is also written text!), participants' accuracy increases. For text, participants did not do well in spotting the fabricated (and non-fabricated) content. Add audio and they score much better. Add video and audio and they score even better. Authors conclude that "Overall, the experiment's results show that as participants have access to more information via audio and video, they are better able to distinguish whether a political speech has been fabricated". But this conclusion is based on an experiment with a very limited scope. The conclusion should be: If you tell participants that half of the stimuli they get displayed is fabricated, then more information via audio and video means better accuracy ratings. Taken together, I think the authors did a good job but the study is more fitting for a journal with a more focused scope.

We are glad R4 enjoyed reading the discussion section, and we include additional experiments per R4's suggestion (and as previously mentioned) to broaden the scope of the experiment by including randomized experiments on not only media modalities but also on audio sources (voice actors and text-to-speech algorithms) and base rates. We now have 4 separate experiments with over 2,000 participants (with an additional 40,000 participants who found the experiments organically) using text, audio, and video from 32 political speeches and video from another 12 videos. We believe this work is particularly relevant for the broad audience of Nature Communications (and not a journal with a more focused scope) because the quality of synthetic media produced by generative AI is increasing at a fast rate, synthetic media is a topic of societal importance and wide concern that impacts a number of fields from psychology to computer science to political science to communications to business to other fields, we offer multiple large scale experiments that offer empirical insights into how people discern between real and fake political speeches across a number of dimensions, which reveals (1) state-of-the-art text-to-speech algorithms are harder to discern than audio produced by voice actors (2) base rates minimally influence discernment of fabricated political speeches (3) human discernment relies more on how something is said, the audio-visual cues, than what is said, the speech content, which adds considerable nuance to the "realism heuristic" in communication theory and research in political science showing that "seeing is believing."

REVIEWER COMMENTS

Reviewer #1 (Remarks to the Author):

Review of NCOMMS-22-37488A-Z

Reviewer: Stephan Lewandowsky

Summary and overall recommendation

This study tested participants' ability to discern real and fake (i.e., manufactured by AI) political speeches, when presented as video, audio, text, or combinations thereof. It further explored the impact of political alignment and cognitive reflection on accurate identification of fake and real speeches. Participants were presented with 32 excerpts of political speeches (half of which were fake) from the Presidential Deepfake Dataset, and were asked to indicate their confidence in whether the speech was fabricated. It was found that participants were more accurate when watching and hearing speeches, compared with just reading a transcript. The authors argue that this reflects a judgement based more on how the politician was speaking (i.e., audio-visual cues) than on what was said. The authors also found that accuracy was moderated by performance on the cognitive reflection task, arguing that a lack of cognitive reasoning leads to over-reliance on content.

I reviewed this paper at the first round (together with a PhD student) and I was positively inclined at the time but withheld endorsement of publication because of a few issues that had to be resolved. In particular, we were concerned about possible training artifacts, and we were concerned about the loose nomenclature. The revision has largely addressed those concerns, and the paper is now in good shape. Nonetheless, some revisions are called for to maximize clarity.

Detailed comments

[page#]:[para#]:[#line]

2:1:1-3 These clauses would be best split into multiple sentences.

3 Figure 1 is never referenced in the text.

6 The bottom panels in Figure 5 would be much easier to read if color coded party (so red for Trump, blue for Biden) rather than status of the video. Similarly, solid vs. broken lines would map nicely into real vs. fake.

6-7 The Results of Experiment 3 are quite difficult to follow: I suspect many readers will skip through this entire section because the unstructured catalogue of effects does not make for pleasant reading. I strongly encourage the authors to find ways of rewriting this to make it more accessible. What is it that's important among all those innumerable t-tests?

7:L:1-2 "increase people's ability" relative to what? This is not clear.

9:3:1 "experiments" not "experiment".

Method: Why is the method written in the present tense?

Reviewer #2 (Remarks to the Author):

The authors report results of a series of experiments investigating human detection of political deepfakes across transcripts, audio, and video. Across these experiments, the key

findings appear to be that 1) misinformation base rates have a minimal impact on detection, 2) deepfakes with state-of-the-art audio generation are harder to detect as fake than those generated with voice actor audio, and 3) audio-visual information in videos aid the detection of deepfakes which is shown by comparing detection of video and text.

I applaud the authors for the efforts taken in generating an improved stimulus set that is more reflective of current state-of-the-art tech capabilities. I also appreciate that the authors have undertaken considerable work in conducting further experiments. I feel that there are some very interesting and important findings in the paper, however, I didn't feel that these findings came through clearly and strongly enough. In my opinion, the results could be more concise to really hammer home the takeout messages, and there should be greater effort to create a story that links the experiments. Currently, the experiments are too isolated from one another without an obvious explanation for why/how each develops upon the previous one. This linkage (and clarity in findings) seems vital for a journal like Nature Communications where a general scientific audience is targeted.

A few more specific comments/queries:

The experimental conditions and results could be more fluent to make sure that the key manipulations and findings are easily digestible by a general scientific audience. Currently, it takes a substantial amount of effort on the reader's part to discern and interpret the important findings. I'd also like to see some consideration of how the experiments develop and build throughout the paper—this narrative is limited which makes the experiments seem isolated rather than all coming together to address an overarching research aim. I appreciate that some experiments might have been conducted in response to previous reviewer comments, however, I don't think it should be too tricky to connect them in a more fluent way.

Figure 5 – it looks like participants are more accurate (except transcript) on fabricated with low rate of fabrications than high rate of fabrications. This finding seems interesting and somewhat counterintuitive. Do you have any theories that might account for this difference in accuracy?

Is the main finding in Expt 4 already stated in Expt 2? I.e., that text-generated voice is harder to detect than actor generated voice? Is the rationale for this study to check that's why this result occurred in Expt 2? If so, it'd be useful to see some rationale for Expt 4 explicitly in the paper.

The discussion opens by stating that "This paper provides evidence, via 4 pre-registered randomized experiments with 2,015 participants that visual and auditory communication modalities increase people's ability to distinguish authentic political speeches from fabricated political speeches." Is this summary a fair reflection of all experiments, for example, what about the base rate manipulation in Experiment 3 or Experiment 4 where the research question focused on detecting whether the audio source was real or a voice actor?

I have a question about the ethics for the experiments. The authors state that "This research

complies with all relevant ethical regulations and the Massachusetts Institute of Technology's Committee on the Use of Humans as Experimental Subjects approved this study as Exempt Category 3 – Benign Behavioral Intervention. This study's exemption identification numbers are E-3105, E-3354, and E-4735."

The use of the phrase "exempt" caught my attention and I thought it important to check if the experimental design and protocol was reviewed by an ethics committee? This could simply be a difference in terminology use across institutions; given the importance of ethics, I'd appreciate the confirmation that the studies were subjected to independent ethical scrutiny though.

Could you confirm that participants were excluded from taking part in more than one of the experiments on Prolific?

The directionality in the questioning is a concern. Please could you explain why the phrase used was "share how confident you are that it is fabricated"/"share whether you think the speech is fabricated" rather than a balanced statement (e.g., share whether you think the speech is real or fabricated)? Did you find a response bias?

With the base rate manipulation, is it that accuracy is simply a reflection of being more likely to respond "fabricated" due to the phrasing of the question?

The concluding statement in the abstract and discussion seem to contradict one another. Please clarify.

Reviewer #3 (Remarks to the Author):

I appreciate the work that the authors have done in responding to my comments. I am happy to endorse publication as this is the most comprehensive examination of how accurately people can detect deepfakes to date that I know of. The editor should confirm that the authors will release the full set of videos, transcripts, etc. as promised because these will be a major contribution to science given how comprehensive they are. My comments follow.

1. I am unclear why Truepic helped pay for the research or what its role in the research was, but I would ask that the editor resolve this issue with the authors before publication.

2. I appreciate the new preregistered studies and the more specific disclosure of what is or isn't preregistered. I hope the authors, editor, and other reviewers will attend to this issue proactively in the future as it is critical to the quality of our science.

3. I appreciate the new studies that vary the base rate of false content without disclosing this variation to participants.

4. For clarity, the authors should clarify *much* sooner that their text-to-speech results are for "text-to-speech audio trained on the speaker," not generic text-to-speech output.

Reviewer #4 (Remarks to the Author):

The authors have put in great effort to expand their study, by adding experiments, focusing also on audio sources, randomizing (and not communicating about) base rates. As such, there is even more to like about this study. This study certainly makes a contribution to the literature. I am still concerned, however, about the magnitude of that contribution.

The issue, in my view, is that in all experiments the participants are forced to consider the veracity of the stimuli. To their credit, the authors recognize this limitation in their discussion. They note: "These experiment are useful to study how people discern multimedia information when attending to questions of accuracy, but they are less useful in understanding how people will share misinformation they consume on social media". That's exactly right. The problem is that people do not process information in a way that comes close to how the participants are forced to process information in this study. This means that the findings are very narrowly applicable. And some conclusions that are being drawn in the paper are too sweeping. For instance the conclusion about what this study adds to the seeing is believing narrative: "people are significantly more accurate at identifying fabricated speeches as fabricated when the speeches include audio and visual modalities as opposed to only text." Yes, but this conclusion can only apply to situations in which people consider accuracy, which they by default rarely do. I know I asked for more data-collection in an earlier review, and I hardly dare to ask, but a data-collection effort in which participants are not as strongly primed to consider the accuracy of the stimuli would turn this very good study into an excellent study. Perhaps an experiment with an unprimed open-ended response option (something like 'please write down what comes to mind after viewing this video') would allow the authors to make stronger claims that are less narrowly applicable.

Responses to R1

Reviewer #1 (Remarks to the Author):

Review of NCOMMS-22-37488A-Z
Reviewer: Stephan Lewandowsky

Summary and overall recommendation

This study tested participants' ability to discern real and fake (i.e., manufactured by AI) political speeches, when presented as video, audio, text, or combinations thereof. It further explored the impact of political alignment and cognitive reflection on accurate identification of fake and real speeches. Participants were presented with 32 excerpts of political speeches (half of which were fake) from the Presidential Deepfake Dataset, and were asked to indicate their confidence in whether the speech was fabricated. It was found that participants were more accurate when watching and hearing speeches, compared with just reading a transcript. The authors argue that this reflects a judgement based more on how the politician was speaking (i.e., audio-visual cues) than on what was said. The authors also found that accuracy was moderated by performance on the cognitive reflection task, arguing that a lack of cognitive reasoning leads to over-reliance on content.

I reviewed this paper at the first round (together with a PhD student) and I was positively inclined at the time but withheld endorsement of publication because of a few issues that had to be resolved. In particular, we were concerned about possible training artifacts, and we were concerned about the loose nomenclature. The revision has largely addressed those concerns, and the paper is now in good shape. Nonetheless, some revisions are called for to maximize clarity.

Thank you for reviewing this paper twice, positively recommending this manuscript, and offering detailed feedback that helped this manuscript maximize clarity.

Detailed comments

[page#]:[para#]:[#line]

2:1:1-3 These clauses would be best split into multiple sentences.

We agree, and we have split these clauses into multiple sentences as follows: "Related research demonstrates how fake images can be persuasive and difficult to distinguish from real images. People rarely question the authenticity of images even when primed. Images can increase the credibility of disinformation. Images of synthetic faces produced by StyleGAN2 are indistinguishable to research participants from the original photos on which the StyleGAN2 algorithm was trained."

3 Figure 1 is never referenced in the text.

Thank you for catching this, and we have now included a reference to Figure 1 as follows: “The stimuli include 32 videos from the Presidential Deepfake Dataset and 12 additional videos (see Figure 1 for a screenshot from each video).”

6 The bottom panels in Figure 5 would be much easier to read if color coded party (so red for Trump, blue for Biden) rather than status of the video. Similarly, solid vs. broken lines would map nicely into real vs. fake.

Thank you for this suggestion! We agree, and we have changed the mapping of colors and dashes in Figure 5 accordingly.

6-7 The Results of Experiment 3 are quite difficult to follow: I suspect many readers will skip through this entire section because the unstructured catalogue of effects does not make for pleasant reading. I strongly encourage the authors to find ways of rewriting this to make it more accessible. What is it that’s important among all those innumerable t-tests?

We revised the paragraph with the pre-registered 6 families of 32 t-tests each to make it more accessible by deleting unnecessary analyses and pointing to the tables of statistical tests from the very beginning. We pre-registered these t-tests because we wanted to demonstrate the robustness of our findings on both the overall and per-video level; these t-tests demonstrate the findings are consistently in the same direction as opposed to heterogeneous depending on the context of the stimuli and we begin the paragraph with this justification, which helps make the reading more clear and pleasant. Thank you!

7:L:1-2 “increase people’s ability” relative to what? This is not clear.

In order to increase clarity, we re-wrote this sentence as “For each additional stimulus seen, we find the participants’ accuracy increases by 0.3 percentage points ($p=0.011$); participants slightly improve based on seeing and hearing real and fabricated stimuli.”

9:3:1 “experiments” not “experiment”.

Thanks for catching this. We fixed the typo.

Method: Why is the method written in the present tense?

We have now revised the methods section to be written in the past tense. Thank you for this catch.

Responses to R2

Reviewer #2 (Remarks to the Author):

The authors report results of a series of experiments investigating human detection of political deepfakes across transcripts, audio, and video. Across these experiments, the key findings appear to be that 1) misinformation base rates have a minimal impact on detection, 2) deepfakes with state-of-the-art audio generation are harder to detect as fake than those generated with voice actor audio, and 3) audio-visual information in videos aid the detection of deepfakes which is shown by comparing detection of video and text.

I applaud the authors for the efforts taken in generating an improved stimulus set that is more reflective of current state-of-the-art tech capabilities. I also appreciate that the authors have undertaken considerable work in conducting further experiments. I feel that there are some very interesting and important findings in the paper, however, I didn't feel that these findings came through clearly and strongly enough. In my opinion, the results could be more concise to really hammer home the takeout messages, and there should be greater effort to create a story that links the experiments. Currently, the experiments are too isolated from one another without an obvious explanation for why/how each develops upon the previous one. This linkage (and clarity in findings) seems vital for a journal like Nature Communications where a general scientific audience is targeted.

Thank you for applauding the improved stimulus set and encouraging more concise and fluent writing to link the experiments. The updated manuscript now more clearly links each experiment to the rest by (1) signposting, (2) clear explanations of the hypothesis that each experiment is testing in the introduction, (3) adding Table 1 to the introduction to clarify what each experiment is, and (4) edits in the results to enhance clarity and ease of reading.

A few more specific comments/queries:

The experimental conditions and results could be more fluent to make sure that the key manipulations and findings are easily digestible by a general scientific audience. Currently, it takes a substantial amount of effort on the reader's part to discern and interpret the important findings. I'd also like to see some consideration of how the experiments develop and build throughout the paper—this narrative is limited which makes the experiments seem isolated rather than all coming together to address an overarching research aim. I appreciate that some experiments might have been conducted in response to previous reviewer comments, however, I don't think it should be too tricky to connect them in a more fluent way.

We edited the results section to include signposting in the titles, contextual threading linking each experiment to the next in the opening sentences in each experiment subsection, and general re-writing for clarity. Thank you for this suggestion, which has made this a much more readable results section.

Figure 5 – it looks like participants are more accurate (except transcript) on fabricated with low rate of fabrications than high rate of fabrications. This finding seems interesting and somewhat counterintuitive. Do you have any theories that might account for this difference in accuracy?

While participants are more accurate (except transcripts) on fabricated in the low base rate condition compared to the high base rate condition, participants are more accurate on transcripts overall in the low rate (60%) than high rate (56%) (and based on a t-test this is statistically significant with $p=0.001$) and we do not find statistically significant differences for overall accuracy in audio, silent video with subtitles, or video with audio. We find that high base

rate leads to higher rate of identifying a stimuli as fabricated, but the higher rate of identifying a stimuli as fabricated isn't enough to compensate for the high base rate. One speculative answer is that the 80% base rate of manipulations was outside of people's expectations and the stimuli were high quality, so participants didn't identify videos as fakes as much as they could/should have whereas a low base rate was perhaps more expected and despite the high quality participants could tell the difference between real videos and the fake stimuli.

Is the main finding in Expt 4 already stated in Expt 2? I.e., that text-generated voice is harder to detect than actor generated voice? Is the rationale for this study to check that's why this result occurred in Expt 2? If so, it'd be useful to see some rationale for Expt 4 explicitly in the paper.

Thank you for raising this question! The short answer is No the main finding in E4 is not in E2, and we have re-written Experiment 4 (and added Table 1 in the Introduction and a description of the experiments in the Introduction) to reduce confusion. In E4, we are evaluating voice actor's voices paired with real videos compared to real speaker's voices in real videos. So, E4 is different than all the rest of the experiments in that there are no visual manipulations in the experiment. We now write, "In order to further identify the role of manipulated audio in participants' ability to distinguish between real and fabricated content across the previous experiments, we conduct Experiment 4... By focusing on only real videos, we examine the role of perfectly realistic visual information to influence accuracy." In addition, we clarify finding in the results: "This suggests the voice actor's audio consistently matched with the real video footage and audio with video increased participants' beliefs that all audio (both real and fake) was authentic relative to when participants were listening to audio only." And we now mention in the discussion: "Experiment 4 offers insights on the future influence of hyper realistic videos by demonstrating that perfectly realistic video (the real videos paired with voice actor audio) leads people to believe audio is more likely to be authentic than when listening to audio by itself."

The discussion opens by stating that "This paper provides evidence, via 4 pre-registered randomized experiments with 2,015 participants that visual and auditory communication modalities increase people's ability to distinguish authentic political speeches from fabricated political speeches." Is this summary a fair reflection of all experiments, for example, what about the base rate manipulation in Experiment 3 or Experiment 4 where the research question focused on detecting whether the audio source was real or a voice actor?

Thank you for raising this concern, and we agree that Experiment 4 is slightly different than Experiments 1-3. We have replaced "4" with multiple, and we have edited the paragraph accordingly to identify the findings from Experiments 1-3 and 5 which focus on the novel contribution around fabricated content and Experiment 4 which focuses only on visually authentic media to evaluate detection of audio manipulations.

I have a question about the ethics for the experiments. The authors state that "This research complies with all relevant ethical regulations and the Massachusetts Institute of Technology's Committee on the Use of Humans as Experimental Subjects approved this study as Exempt Category 3 – Benign Behavioral Intervention. This study's exemption identification numbers are E-3105, E-3354, and E-4735."

The use of the phrase “exempt” caught my attention and I thought it important to check if the experimental design and protocol was reviewed by an ethics committee? This could simply be a difference in terminology use across institutions; given the importance of ethics, I’d appreciate the confirmation that the studies were subjected to independent ethical scrutiny though.

Thank you for raising this point, and yes, these studies were subjected to independent ethical scrutiny. MIT’s institutional review board (IRB) is called Committee on the Use of Humans as Experimental Subjects (COUHES), and we submitted the studies to COUHES, which approved these studies with the designation Exempt Category 3 – Benign Behavioral Intervention. According to MIT, “Exempt status research is excluded from the requirements of the Regulations and from policies dictated by the Common Rule. Exempt status research must however, comply with COUHES Policy, as outlined in the Investigator Responsibilities for Exempt Research.” Moreover, “exempt status” is common at many institutions. For example, Northwestern University describes Exempt status as “Exempt human subjects research is a specific sub-set of “research involving human subjects” that does not require ongoing IRB oversight. Research can qualify for an exemption if it is no more than minimal risk and all of the research procedures fit within one or more of the exemption categories in the federal IRB regulations. Studies that qualify for exemption must be submitted to the IRB for review before starting the research.”

Could you confirm that participants were excluded from taking part in more than one of the experiments on Prolific?

Confirmed for 99.5% of participants, and we now write: “We exclude participants from participating in multiple experiments. However, 9 participants who participated in Experiment 2 also participated in Experiment 4. The results in Experiment 4 are robust to both including and excluding these 9 participants.”

The directionality in the questioning is a concern. Please could you explain why the phrase used was “share how confident you are that it is fabricated”/“share whether you think the speech is fabricated” rather than a balanced statement (e.g., share whether you think the speech is real or fabricated)? Did you find a response bias?

With the base rate manipulation, is it that accuracy is simply a reflection of being more likely to respond “fabricated” due to the phrasing of the question?

In experiments 2, 3, and 4, we adapted the question wording to what you see in Figure 9 (Figure 8 shows the wording/interface for experiment 1) based on feedback from reviewer 1 (Stephan Lewandowsky) that I am quoting here: “However, in future studies it might be better to use more neutral phrasing, such as “...share whether you think the video is fabricated, and how confident you are in this judgment.” We re-worded the experiment with “Please [read/listen/watch] this [transcript/audio clip/video] from [speaker] and share whether you think the speech is fabricated, and how confident you are in this judgment” because (1) it’s more concise than “real or fabricated” (2) real is logically implied given that it’s a binary task as evidenced by the two radio buttons that participants could choose from (where real is on top/first) and the top of the website says “Fabricated or Authentic?”.

We do not find a response bias. In fact in Experiment 3 with the base rate manipulation, we find that the that the base rate of participants identifying stimuli as fake is 55% in the high fake condition and 41% in the low fake condition, which means the low base rate influenced people to move more from the 50-50 mark than the high base rate condition. Moreover, the phrasing is kept consistent over randomly assigned media modalities of speeches, so this cannot be simply a reflection of the phrasing.

We conducted an additional experiment (Experiment 5) with wording that neither has directionality of a question nor direct ask of our dependent variable of interest per request of reviewer 4, and we find the same results continue to hold.

The concluding statement in the abstract and discussion seem to contradict one another. Please clarify.

We have re-written the last statement in the discussion to increase clarity and avoid any potential contradiction with the statement in the abstract: "Finally, these findings offer insights into political communication and communication theory more generally; there is more to how humans form beliefs than the "seeing is believing" narrative would suggest because people can pay attention and seek out inconsistencies in both what is said and how something is said."

Responses to R3

Reviewer #3 (Remarks to the Author):

I appreciate the work that the authors have done in responding to my comments. I am happy to endorse publication as this is the most comprehensive examination of how accurately people can detect deepfakes to date that I know of. The editor should confirm that the authors will release the full set of videos, transcripts, etc. as promised because these will be a major contribution to science given how comprehensive they are. My comments follow.

Thank you for your comments and your enthusiastic endorsement of publication. We include all stimuli (and code and data) for this experiment on Research Box at https://researchbox.org/1723&PEER_REVIEW_passcode=EGVULE, and we plan to remove the peer review passcode upon acceptance so anyone can access the stimuli and data.

1. I am unclear why Truepic helped pay for the research or what its role in the research was, but I would ask that the editor resolve this issue with the authors before publication.

In the competing interests section, we wrote, "The authors declare funding for participant recruitment from Truepic and otherwise declare no competing interests." For context, we note that the MIT Media Lab is funded by a consortium of around 50-80 companies in addition to individual donors and government grants. Truepic funded participant recruitment, which totaled \$7,000, because Truepic was interested in supporting research on how people detect deepfakes and I (Matt Groh) requested the funding once I calculated the budget required for conducting

additional experiments requested by reviewers. No one on the team has accepted any additional funding from Truepic. Moreover, we worked with Truepic to integrate C2PA metadata into the deepfakes because Truepic had the only available technology to seamlessly integrate this provenance metadata into the videos that we used for clear and simple debriefing of which videos are real and fake to participants at the end of the experiment.

2. I appreciate the new preregistered studies and the more specific disclosure of what is or isn't preregistered. I hope the authors, editor, and other reviewers will attend to this issue proactively in the future as it is critical to the quality of our science.

We appreciate your attention to these details and we agree!

3. I appreciate the new studies that vary the base rate of false content without disclosing this variation to participants.

Thank you and we appreciate the reviewers suggestions to include this.

4. For clarity, the authors should clarify *much* sooner that their text-to-speech results are for "text-to-speech audio trained on the speaker," not generic text-to-speech output.

Thank you for this comment. We've edited the manuscript at the first mention of "text-to-speech" in the paper to include "text-to-speech algorithm fine-tuned on the presidents' voices".

Responses to R4

The authors have put in great effort to expand their study, by adding experiments, focusing also on audio sources, randomizing (and not communicating about) base rates. As such, there is even more to like about this study. This study certainly makes a contribution to the literature. I am still concerned, however, about the magnitude of that contribution.

The issue, in my view, is that in all experiments the participants are forced to consider the veracity of the stimuli. To their credit, the authors recognize this limitation in their discussion. They note: "These experiment are useful to study how people discern multimedia information when attending to questions of accuracy, but they are less useful in understanding how people will share misinformation they consume on social media". That's exactly right. The problem is that people do not process information in a way that comes close to how the participants are forced to process information in this study. This means that the findings are very narrowly applicable. And some conclusions that are being drawn in the paper are too sweeping. For instance the conclusion about what this study adds to the seeing is believing narrative: "people are significantly more accurate at identifying fabricated speeches as fabricated when the speeches include audio and visual modalities as opposed to only text." Yes, but this conclusion can only apply to situations in which people consider accuracy, which they by default rarely do. I know I asked for more data-collection in an earlier review, and I hardly dare to ask, but a data-collection effort in which participants are not as strongly primed to consider the accuracy of the stimuli would turn this very good study into an excellent study. Perhaps an experiment with an unprimed open-ended response option (something like 'please write down what comes to

mind after viewing this video') would allow the authors to make stronger claims that are less narrowly applicable.

Thank you for recognizing the revised and expanded study as a clear contribution to the literature and re-raising the issue of the role of attention that we discuss as a limitation in the discussion. We deeply appreciate your encouragement to “turn this very good study into an excellent study” by conducting an additional experiment with “an unprimed open-ended response option”, and we have done just that with Experiment 5. In particular, we based the wording of the question in Experiment 5 on your suggested wording, and we express it as “What comes to mind after watching the following video/listening to the following audio/reading the following quote?” depending on the media modality shown to the participant. We now include this additional pre-registered experiment in the paper (see Figure 7 and Table 18 for detailed results), and we find results in line with the rest of the experiments, which as you suggest allow stronger claims that are less narrowly applicable. As such, we update the discussion. Thank you!

REVIEWERS' COMMENTS

Reviewer #1 (Remarks to the Author):

Review of NCOMMS-22-37488B

Reviewer: Stephan Lewandowsky and Simon Clark

We reviewed the paper twice before and were pleased by the positive trajectory exhibited on both occasions. This revision has addressed concerns raised at the two previous rounds and we are now happy to recommend publication. We are confident that this paper will make a nice contribution.

Reviewer #2 (Remarks to the Author):

Thanks for the responses to my previously raised points. I do not have any additional comments to add.

Reviewer #3 (Remarks to the Author):

I already supported publication of this manuscript, but the changes make it even better. It's clearer and easier to follow and Study 5 is a notable addition to the provided evidence.

Here are two small issues I noticed while reading:

1. The authors providing the range and the maximum Kappa values is inadequate. They should provide the full set of Kappa values for each possible pairing of coders as well as the relevant combined value for the dataset as a whole.
2. Qualtrics is erroneously spelled with a lower-case q in one case.

Reviewer #4 (Remarks to the Author):

The authors have addressed all my points more than sufficiently. They have conducted rigorous studies, and present valuable and fascinating insights. I would like to endorse this manuscript for publication.

Reviewer #4 (Remarks on code availability):

Clear, organized and reproducible.